# Meta-Exploiting Frequency Prior for Cross-Domain Few-Shot Learning

**Fei Zhou[1]    Peng Wang[2]    Lei Zhang[1]\*    Zhenghua Chen[3]    Wei Wei[1]**
**Chen Ding[4]    Guosheng Lin[5]    Yanning Zhang[1]**

[1] School of Computer Science, Northwestern Polytechnical University
[2] School of Computer Science and Engineering, University of Electronic Science and Technology of China
[3] Institute for Infocomm Research, and Centre for Frontier AI Research, A\*STAR
[4] School of Computer Science, Xi'an University of Posts & Telecommunications
[5] School of Computer Science and Engineering, Nanyang Technological University

zhoufei@mail.nwpu.edu.cn   wangpeng8619@gmail.com
chen0832@e.ntu.edu.sg   gslin@ntu.edu.sg   dingchen@xupt.edu.cn
{nwpuzhanglei,weiweinwpu,ynzhang}@nwpu.edu.cn

## Abstract

Meta-learning offers a promising avenue for few-shot learning (FSL), enabling models to glean a generalizable feature embedding through episodic training on synthetic FSL tasks in a source domain. Yet, in practical scenarios where the target task diverges from that in the source domain, meta-learning based method is susceptible to over-fitting. To overcome this, we introduce a novel framework, Meta-Exploiting Frequency Prior for Cross-Domain Few-Shot Learning, which is crafted to comprehensively exploit the cross-domain transferable image prior that each image can be decomposed into complementary low-frequency content details and high-frequency robust structural characteristics. Motivated by this insight, we propose to decompose each query image into its high-frequency and low-frequency components, and parallel incorporate them into the feature embedding network to enhance the final category prediction. More importantly, we introduce a feature reconstruction prior and a prediction consistency prior to separately encourage the consistency of the intermediate feature as well as the final category prediction between the original query image and its decomposed frequency components. This allows for collectively guiding the network's meta-learning process with the aim of learning generalizable image feature embeddings, while not introducing any extra computational cost in the inference phase. Our framework establishes new state-of-the-art results on multiple cross-domain few-shot learning benchmarks.

## 1 Introduction

Meta-learning Finn et al. [2017], Lee et al. [2019], Rusu et al. [2019], Zhmoginov et al. [2022], Zhang et al. [2023], Baik et al. [2020] represents a potent paradigm within the domain of FSL Vinyals et al. [2016], Snell et al. [2017], Huang et al. [2022], Zhang and Huang [2022], Chen et al. [2021]. This paradigm harnesses a feature embedding network to capture task-agnostic meta-knowledge, facilitating generalization to novel tasks. To this end, meta-learning systematically samples a sequence of FSL episodes in the source domain to supervisedly enforcing learn an effective feature embedding network that assimilating cross-task transferable essentials and generalize well to novel target tasks. Due to its exceptional learning-to-learn capabilities, meta-learning has established itself as the de facto approach for the development of effective few-shot solvers Vinyals et al. [2016], Snell et al.

---

\*Corresponding author.

[2017], Huang et al. [2022], Zhang and Huang [2022], Finn et al. [2017], Lee et al. [2019], Rusu et al. [2019], Zhmoginov et al. [2022], Zhang et al. [2002], Baik et al. [2020].

However, in practical cross-domain scenarios where the target task exhibits a noticeable distribution discrepancy from that in the source domain, meta-learning based methods are susceptible to over-fitting. This phenomenon can be attributed to two main reasons. Firstly, tasks randomly sampled in source domain often come from one or several fixed patterns, and thus the continual switching of episodes training may cause the model to over-fit on some task-specific priors. For instance, in tasks involving the discrimination between tigers and giraffes, meta-learning methods may compel the model to emphasize appearance outlines, while in tasks focused on fine-grained bird identification, models tend to prioritize local discriminative textures. Yet, these task-specific priors prove challenging to transfer across different tasks Lyu et al. [2021], Zhou et al. [2023]. Secondly, the iterative episodic training in the source domain can result in the model over-fitting to semantic prior properties specific to that domain. For example, source domains comprised of natural scene images often exhibit obvious semantic priors, whereas specialized target domains like medical image analysis or remote sensing may lack clear semantic concepts. This overall domain bias also hampers the model's cross-domain generalization. For all these challenges, the underlying evil lies in the absence of cross-domain invariant priors to guide meta-learning in the source domain.

To address this challenge, we introduce a novel framework, Meta-Exploiting Frequency Prior for Cross-Domain Few-Shot Learning. Inspired by classical image transform theories (Fourier Nuss-baumer and Nussbaumer [1982] or wavelet Zhang and Zhang [2019]), where each image can be decomposed into low-frequency content details and high-frequency structural characteristics, despite which domain it belongs to, we attempt to cast such a cross-domain invariant image property into appropriate frequency priors and utilize them to guide the meta-learning in source domain. Following this idea, we decompose each query image into a high-frequency and a low-frequency parts, and feed each into the feature embedding network for final category prediction, mirroring the process applied to the original query image. These allows for the independent feature learning in both spatial and frequency domains. In addition, the low-frequency and high-frequency branch will separately exploit the complementary image content and structures for feature enhancement, which are often concealed in the spatial domain of original query image. More importantly, we further develop two frequency priors, namely a feature reconstruction prior and a prediction consistency prior, which separately forces the original query image and its decomposed frequency components to produce the consistent intermediate feature representation as well as the final category prediction. In a specific, the feature reconstruction prior requires to reconstruct the feature of original image through fusing the features of both decomposed frequency parts using a deep projection network. The prediction consistency prior aims to minimize the separate Kullback-Leibler divergence between the prediction scores produced by the original query image and its each frequency component. By doing these, meta-learning in the source domain can be appropriately regularized and produce the exceptional cross-domain generalizable feature embeddings. Moreover, such frequency priors only perform in the meta-learning phase without introducing any extra computational cost in inference. Through a series of rigorous experiments, our framework establishes itself as a front-runner, achieving state-of-the-art results across multiple cross domain FSL benchmarks. Additionally, our method exhibits significant efficiency advantages.

The primary contributions of this study can be summarized as follows:

- We present a novel insightful meta-learning framework that exploits cross-domain invariant frequency priors to alleviate the over-fitting problems of classic meta-learning in cross-domain FSL tasks.
- We propose two frequency prior, namely a prediction consistency prior and a feature reconstruction prior, to collectively guide the meta-learning procedure.
- We achieve state-of-the-art results on multiple cross-domain FSL benchmarks.

## 2  Methodology

**Problem formulation.**  Cross-Domain Few-Shot Learning (CD-FSL) aims to transfer the knowledge acquired by a model in the source domain $\mathcal{D}_s$ to perform few-shot tasks in the target domain $\mathcal{D}_t$. It is noteworthy that the categories in $\mathcal{D}_t$ differ from those in the source domain. Each task $\mathcal{T}$ involves the random sampling of $N$ categories, with $K$ samples and $M$ samples randomly selected

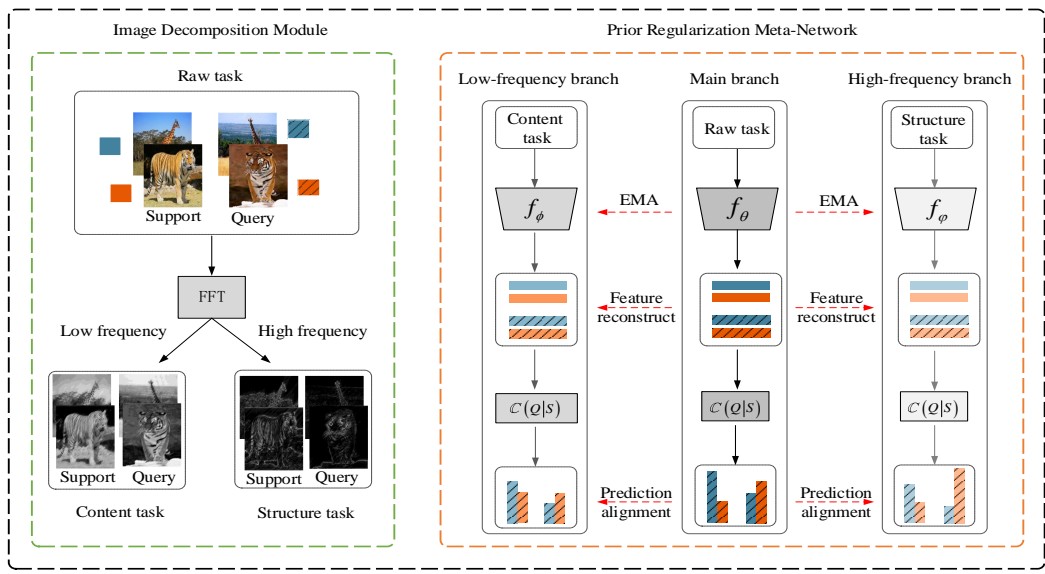

Figure 1: Framework of the proposed method. In this work, we present an insightful meta-learning framework that exploits cross-domain invariant frequency priors to alleviate the over-fitting problems of classic meta-learning in cross-domain FSL tasks. Our method consists of an Image Decomposition Module (IDM) and a Prior Regularization Meta-Network (PRM-Net). Among them, IDM aim at explicitly decomposing every image in few-shot task into low- and high-frequency components. PRM-Net develops a prediction consistency prior and a feature reconstruction prior to jointly regularize the feature embedding network during meta-learning, aiming to learn generalizable image feature embeddings. Once the model is trained, only the main branch is retained for meta-testing on target domains.

from each category to constitute the support set $\mathcal{T}_S$ and the query set $\mathcal{T}_Q$, respectively. The support set $\mathcal{T}_S$ is employed for constructing a task-specific classifier, while the query set $\mathcal{T}_Q$ is used to assess the classification accuracy for that specific task. To emulate the meta-testing process, methods based on meta-learning typically sample a series of few-shot tasks from the source domain for training.

**Overview.** In this study, we introduce a sophisticated meta-learning framework that leverages cross-domain invariant frequency priors to mitigate the over-fitting problems of classic meta-learning in cross-domain FSL tasks. As illustrated in Fig. 1, our method comprises two key components: the Image Decomposition Module (IDM) and the Prior Regularization Meta-Network (PRM-Net). The IDM is designed to explicitly decompose each image within a few-shot task into its low- and high-frequency components using Fast Fourier Transform (FFT) Nussbaumer and Nussbaumer [1982]. PRM-Net is a key component responsible for introducing a prediction consistency prior and a feature reconstruction prior. PRM-Net is organized into three branches: the main branch, the low-frequency content branch, and the high-frequency structure branch. In each branch, all images undergo feature extraction through the embedding network. Subsequently, a task-specific classifier is constructed based on the support set to predict the query set. Two frequency priors, namely the prediction consistency prior and the feature reconstruction prior, are proposed to collectively guide the network's meta-learning process with the aim of learning generalizable image feature embeddings. The IDM and PRM-Net work collaboratively to provide a robust meta-learning framework, aiming to enhance cross-domain generalization by explicitly considering image decomposition and introducing effective regularization during the meta-learning process. The subsequent sections will provide a detailed description of each module.

## 2.1 Image Decomposition Module

In the realm of signal processing, classical image transform theory Nussbaumer and Nussbaumer [1982], Zhang and Zhang [2019] posits that every image can be decomposed into low-frequency content and high-frequency structure, irrespective of its domain. Therefore, within the Image Decom-

position Module, we adhere to this theory and employ Fast Fourier Transform (FFT) Nussbaumer and Nussbaumer [1982] to explicitly decompose each image from the few-shot task $\mathcal{T}$ into a low-frequency content image and a high-frequency structure image. Specifically, for an image $x$ in $\mathcal{T}$, the initial step involves decomposing it into the frequency domain:

$$\left[ f^{x^{low}}, f^{x^{high}} \right] = \mathcal{F}(x), \tag{1}$$

where $\mathcal{F}$ represents FFT, $f^{x^{low}}$ and $f^{x^{high}}$ represent the low-frequency and high-frequency components of $x$ in the frequency domain respectively. Following this decomposition, we transform these components back into the image space using the inverse FFT:

$$\begin{aligned} x^{low} &= \mathcal{F}^{-1}\left( f^{x^{low}} \right), \\ x^{high} &= \mathcal{F}^{-1}\left( f^{x^{high}} \right), \end{aligned} \tag{2}$$

where $\mathcal{F}^{-1}$ represents inverse transform of FFT, $x^{low}$ and $x^{high}$ represent the decomposed low-frequency content image and high-frequency structure image respectively. Similarly, we apply the same decomposition process to each image in $\mathcal{T}$ to obtain the corresponding low-frequency content task $\mathcal{T}^{low}$ and high-frequency structure task $\mathcal{T}^{high}$.

## 2.2 Prior Regularization Meta-Network

The proposed Prior Regularization Meta-Network is designed to leverage cross-domain invariant frequency priors, addressing meta-learning over-fitting in the source domain. To achieve this objective, we introduce a three-branch meta-learning network, dedicated to processing the raw few-shot task $\mathcal{T}$, the low-frequency task $\mathcal{T}^{low}$, and the high-frequency task $\mathcal{T}^{high}$, respectively. Significantly, we propose a prediction consistency prior and a feature reconstruction prior to jointly regularize the feature embedding network during meta-learning. This approach empowers the learning process, facilitating the acquisition of a cross-domain generalizable feature embedding. Upon completing the meta-training on the source domain, we discard the high-frequency and low-frequency branches, retaining only the main branch for cross-domain validation.

**The main branch.** As depicted in Fig. 1, the main branch includes a feature embedding network and a task-specific classifier. For a few-shot task $\mathcal{T}$, the main branch first feeds each image into the feature embedding network to obtain features, and then utilizes the support set $\mathcal{T}_S$ to build a prototype classifier Snell et al. [2017]:

$$c_n = \frac{1}{K} \sum_{k=1}^{K} f_\theta(x_{n,k}), \tag{3}$$

where $c_n$ represents the prototype of the $n$-th category, $x_{n,k}$ represents the $k$-th support sample of the $n$-th category, $f_\theta$ represents the feature embedding network. Finally, we utilize the prototype classifier to make prediction on the query set:

$$\mathcal{P}_{x_j} = \frac{\exp\left(-d\left(f_\theta(x_j), c_n\right)\right)}{\sum_{n'} \exp\left(-d\left(f_\theta(x_j), c_{n'}\right)\right)}, n \in [1, N], \tag{4}$$

where $x_j \in \mathcal{T}_Q$, $\mathcal{P}_{x_j}$ represents the prediction scores of $x_j$, $d(\cdot)$ represents the Euclidean distance. For the query image $x_j$, the category corresponding to the highest score in $\mathcal{P}_{x_j}$ is used as the predicted label $\hat{y}_{x_j}$. Subsequently, we calculate the cross-entropy loss between the predicted label $\hat{y}_{x_j}$ and the ground truth $y_{x_j}$ as:

$$\mathcal{L}_{x_j}^{ce} = H(\hat{y}_{x_j}, y_{x_j}), \tag{5}$$

where $H(\cdot)$ denotes the cross-entropy loss function.

**The high- or low-frequency branch.** As illustrated in Fig. 1, both the high-frequency branch and the low-frequency branch maintain consistency with the architecture of the main branch. In practice, we input the decomposed high-frequency task $\mathcal{T}^{high}$ and low-frequency task $\mathcal{T}^{low}$ into these two branches, respectively, to obtain the corresponding features and prediction scores for the query set. Mathematically, the prediction scores for the query image $x_j$ in the high-frequency branch and the low-frequency branch are denoted as $\mathcal{P}_{x_j}^{low}$ and $\mathcal{P}_{x_j}^{high}$, respectively.

**Frequency prior regularization.** In this work, we posit that the over-fitting problem is the core crux that limits the cross-domain generalization of meta-learning model. To this end, we resort to cross-domain invariant priors to regularize meta-learning in the source domain. Motivated by this perspective, we propose a prediction consistency prior and a feature reconstruction prior to jointly regularize the feature embedding network during meta-learning using high-low frequency information obtained from image decomposition. Specifically, the prediction consistency prior aims to minimize the separate Kullback-Leibler divergence between the prediction scores produced by the original query image and its each frequency component. Formally, for a query image $x_j$, we align its high-frequency prediction distribution $\mathcal{P}_{x_j}^{high}$ and low-frequency prediction distribution $\mathcal{P}_{x_j}^{low}$ with the original distribution $\mathcal{P}_{x_j}$ respectively:

$$\mathcal{L}_{x_j}^{align} = D_{KL}\left(\mathcal{P}_{x_j}^{low}||\mathcal{P}_{x_j}\right) + D_{KL}\left(\mathcal{P}_{x_j}^{high}||\mathcal{P}_{x_j}\right), \tag{6}$$

where $D_{KL}\left(\cdot\right)$ is the Kullback-Leibler divergence loss function. The rationale behind this approach is twofold. Firstly, through explicit decomposition-alignment, we compel the model to attend to both low-frequency content and high-frequency structure. Despite their distinct nature, these two types of features synergistically contribute and complement each other in the challenge of cross-domain generalization. Secondly, establishing prediction consistency between high-low frequency and the original one is domain-invariant. This consistency aids the model in generalizing effectively across different domains.

The feature reconstruction prior aims at reconstructing the original features utilizing low-frequency and high-frequency information in the latent space, which promotes the model to learn comprehensive representations. Specifically, we first project embedding features into the low-dimensional latent space, and then utilize the information retained by high- and low-frequency to reconstruct the original features:

$$z_{x_i} = g_\eta\left(f_\theta\left(x_i\right)\right), \tag{7}$$

$$\hat{z}_{x_i} = g_\eta\left(f_\phi\left(x_i^{low}\right)\right) + g_\eta\left(f_\varphi\left(x_i^{high}\right)\right), \tag{8}$$

where $f_\phi$ and $f_\varphi$ are the feature embedding network of the low-frequency branch and the high-frequency branch respectively, $g_\eta$ is a projector composed of one layer full connected neural network (512×256), $\hat{z}_{x_i}$ is the reconstructed feature. Then, the reconstruction loss is calculated as:

$$\mathcal{L}_{x_i}^{recon} = MSE\left(\hat{z}_{x_i}, z_{x_i}\right), \tag{9}$$

where $MSE\left(\cdot\right)$ represents the mean square error loss function.

**Meta-training.** Based on the description provided above, for a few-shot task $\mathcal{T}$, the total loss can be formulated as:

$$\mathcal{L} = \frac{1}{|\mathcal{T}_Q|}\sum\nolimits_{x_j \in \mathcal{T}_Q}\left(\mathcal{L}_{x_j}^{ce} + \mathcal{L}_{x_j}^{align}\right) + \frac{1}{|\mathcal{T}|}\sum\nolimits_{x_i \in \mathcal{T}}\mathcal{L}_{x_i}^{recon}. \tag{10}$$

Following this, we compute the gradient based on the total loss $\mathcal{L}$ to update both the main branch $\theta$ and the projector $\eta$. While one straightforward approach is to share parameters between the high-low frequency branches and the main branch, this might lead the feature embedding network to primarily focus on common features among the three, potentially causing distinctive features in the high-frequency or low-frequency branches to be overlooked. To address this concern and extract more distinctive features, we opt for an explicit design where three separate feature embedding networks are employed without parameter sharing. However, updating the high-frequency and low-frequency branches through gradient back-propagation can introduce additional computational overhead. As a solution, we update the high-frequency branch $\varphi$ and low-frequency branch $\phi$ as the Exponential Moving Average (EMA) of the main branch $\theta$ during meta-training:

$$\begin{aligned} \phi &\leftarrow m_1\phi + \left(1 - m_1\right)\theta, \\ \varphi &\leftarrow m_2\varphi + \left(1 - m_2\right)\theta, \end{aligned} \tag{11}$$

where $m_1$ and $m_2$ are momentum hyper-parameters. We describe the entire meta-training process in detail in Algorithm 1.

---
**Algorithm 1:** Meta-training algorithm of the proposed method.

---
**Input:** Source domain $\mathcal{D}_s$, main branch $f_\theta$, low-frequency branch $f_\phi$, high-frequency branch $f_\varphi$, projector $g_\eta$

**while** *not converged* **do**

    1. Sample a few-shot task $\mathcal{T} = \{\mathcal{T}_S, \mathcal{T}_Q\}$ from $\mathcal{D}_s$;

    **for** *each image $x_i$ in $\mathcal{T}$* **do**

        2. Decompose $x_i$ to obtain high-frequency image $x_i^{high}$ and low-frequency image $x_i^{low}$;

        3. Utilize $f_\theta$, $f_\varphi$ and $f_\phi$ to extract feature for $x_i$, $x_i^{high}$ and $x_i^{low}$ respectively;

        4. Calculate the reconstruction loss $\mathcal{L}_{x_i}^{recon}$ according to Eq. 7, Eq. 8 and Eq. 9;

    5. Build a prototype classifier for each branch separately based on the $\mathcal{T}_S$ in each branch;

    **for** *each query image $x_j$ in $\mathcal{T}_Q$* **do**

        6. Calculate the prediction scores $\mathcal{P}_{x_j}$, $\mathcal{P}_{x_j}^{high}$ and $\mathcal{P}_{x_j}^{low}$ according to Eq. 4;

        7. Calculate the cross-entropy loss $\mathcal{L}_{x_j}^{ce}$ and alignment loss $\mathcal{L}_{x_j}^{align}$ according to Eq. 5 and Eq. 6 respectively;

    8. Calculate the total loss $\mathcal{L}$ according to Eq. 10, and update $\theta$ and $\eta$ via gradient backpropagation;

    9. Update $\varphi$ and $\phi$ according to Eq. 11.

**Output:** The main branch $f_\theta$.

---

**Cross-domain evaluation.** Once the model is trained, only the main branch $f_\theta$ is retained for meta-testing on target domains. The proposed method is designed to enable the model to learn cross-domain transferable knowledge during the training phase, achieving effective generalization without relying on task-level feature extractor fine-tuning during meta-testing. Specifically, for each meta-testing task in the target domain, the main branch is utilized to extract features. Subsequently, the support set $\mathcal{T}_S$ is employed to construct a task-specific classifier for inference on the query set $\mathcal{T}_Q$. It's important to note that the proposed method does not require image decomposition during the meta-testing phase, thereby avoiding additional computational overhead.

## 3 Experimental Analysis

In this section, we begin by providing a detailed description of the experimental configuration, encompassing pre-training, meta-training, and meta-testing. Following that, we analyze the advantages of the proposed method in comparison with state-of-the-art methods. Lastly, we delve into a comprehensive ablation study to further investigate the effectiveness of our approach. Due to space limitations, we put more experiments and analyses in the appendix.

### 3.1 Experimental details

**Source domain and target domains.** We focus on the most challenging scenario of single-source domain Cross-Domain Few-Shot Learning (CD-FSL). Following the established setup Guo et al. [2020], Li et al. [2022], Zhou et al. [2023], we employ the base classes of the *mini*-ImageNet Vinyals et al. [2016] as the source domain dataset. Our model is evaluated across multiple target domains, encompassing natural image domains (*CUB*, *Cars*, *Places*, *Plantae*), remote sensing domain (*EuroSAT*), agricultural domain (*CropDisease*), and medical domains (*ChestX*, *ISIC*). These datasets are widely recognized in the field of cross-domain few-shot learning. Additional details about each dataset can be found in Guo et al. [2020], Tseng et al. [2019].

**Pre-training and meta-training.** In the context of CD-FSL, pre-training is a common technique Li et al. [2022], Zhou et al. [2023], Hu and Ma [2022], Wang and Deng [2021], aiming to provide feature initialization for meta-training. Specifically, it involves supervised classification on the source domain through batch training. Following Li et al. [2022], Zhou et al. [2023], Hu and Ma [2022], we utilize ResNet-10 as the feature embedding network and a one-layer fully connected neural network as the classifier. The total number of pre-training epochs is set to 400. After pre-training, only the feature embedding network is retained as the feature extractor for meta-training. During the meta-training phase, we employ Adam as the optimizer and conduct meta-training for 50 epochs with

Table 1: Comparison with state-of-the-art methods on 5-way 1-shot cross-domain FSL. Average classification accuracies (%) are provided. $\dagger$ stands for exploiting the full data of FSL task. $*$ means that the feature embedding network needs to be fine-tuned (Ft) on each target domain tasks. The best results are in bold.

| Methods | Ft | CUB | Cars | Places | Plantae | Chest | ISIC | EuroSAT | CropDisease | Ave. |
|---|---|---|---|---|---|---|---|---|---|---|
| MatchingNet Vinyals et al. [2016] | ✗ | 35.89 | 30.77 | 49.86 | 32.70 | 20.91 | 29.46 | 50.67 | 48.47 | 37.34 |
| RelationNet Sung et al. [2018] | ✗ | 41.27 | 30.09 | 48.16 | 31.23 | 21.95 | 30.53 | 49.08 | 53.58 | 38.24 |
| GNN Garcia and Bruna [2018] | ✗ | 44.40 | 31.72 | 52.42 | 33.60 | 21.94 | 30.14 | 54.61 | 59.19 | 41.00 |
| FWT Tseng et al. [2019] | ✗ | 45.50 | 32.25 | 53.44 | 32.56 | 22.00 | 30.22 | 55.53 | 60.74 | 41.53 |
| LRP Sun et al. [2021] | ✗ | 48.29 | 32.78 | **54.83** | 37.49 | 22.11 | 30.94 | 54.99 | 59.23 | 42.58 |
| ATA Wang and Deng [2021] | ✗ | 45.00 | 33.61 | 53.57 | 34.42 | 22.10 | 33.21 | 61.35 | 67.47 | 43.84 |
| AFA Hu and Ma [2022] | ✗ | 46.86 | 34.25 | 54.04 | 36.76 | 22.92 | 33.21 | 63.12 | 67.61 | 44.85 |
| LDP-net Zhou et al. [2023] | ✗ | 49.82 | 35.51 | 53.82 | 39.84 | **23.01** | 33.97 | **65.11** | 69.64 | 46.34 |
| Ours | ✗ | **51.55** | **37.04** | 52.06 | **41.55** | 22.82 | **33.98** | 64.31 | **71.47** | **46.85** |
| ATA$^\dagger$ Wang and Deng [2021] | ✗ | 50.26 | 34.18 | 57.03 | 39.83 | 21.67 | **34.70** | 65.94 | 77.82 | 47.68 |
| AFA$^\dagger$ Hu and Ma [2022] | ✗ | 50.85 | 38.43 | 60.29 | 40.27 | 21.69 | 34.25 | 66.17 | 72.44 | 48.05 |
| RDC$^\dagger$ Li et al. [2022] | ✗ | 47.77 | 38.74 | 58.82 | 41.88 | **22.66** | 32.29 | 67.58 | 80.88 | 48.83 |
| GNN+wave-SAN$^\dagger$ Fu et al. [2022] | ✗ | 50.25 | 33.55 | 57.75 | 40.71 | 22.93 | 33.35 | 69.64 | 70.80 | 47.37 |
| LDP-net$^\dagger$ Zhou et al. [2023] | ✗ | 55.94 | 37.44 | 62.21 | 41.04 | 22.21 | 33.44 | **73.25** | 81.24 | 50.85 |
| StyleAdv$^\dagger$ Fu et al. [2023] | ✗ | 48.49 | 34.64 | 58.58 | 41.13 | 22.64 | 33.96 | 70.94 | 74.13 | 48.06 |
| Ours$^\dagger$ | ✗ | **59.48** | **38.86** | **62.90** | **44.06** | 22.48 | 34.28 | 69.56 | **84.01** | **51.95** |
| Fine-tuning$^*$ Guo et al. [2020] | ✔ | 43.53 | 35.12 | 50.57 | 38.77 | 22.13 | 34.60 | 66.17 | 73.43 | 45.54 |
| ATA$^{*\dagger}$ Wang and Deng [2021] | ✔ | 51.89 | 38.07 | 57.26 | 40.75 | 22.45 | 35.55 | 70.84 | 82.47 | 49.91 |
| RDC$^{*\dagger}$ Li et al. [2022] | ✔ | 50.09 | 39.04 | 61.17 | 41.30 | 22.32 | 36.28 | 70.51 | 85.79 | 50.81 |

a learning rate of 0.001. In each epoch, we randomly sample 100 meta-tasks, where each meta-task consists of 5-way 5-shot 15-query. Data augmentation techniques such as "Resize," "ImageJitter," and "RandomHorizontalFlip" are applied during meta-training. We set hyper-parameters $m_1$=0.997 and $m_2$=0.999. All experiments were performed on a 4090 GPU. Our experimental platform is a 4090 GPU. Further details and verification of hyper-parameters can be found in the supplementary material.

**Meta-testing.** Upon completion of meta-training, we directly employ the learned model for meta-testing across all target domains. Specifically, for each target domain, we randomly sample 600 meta-tasks for testing. We consider two challenging meta-tasks: a 5-way 1-shot 15-query task and a 5-way 5-shot 15-query task. In each meta-task, we learn a Logistic Regression classifier using the support set and then conduct inference on the query set.

### 3.2 Comparison with state-of-the-art methods

**Methods.** In the realm of single-source domain CD-FSL, the state-of-the-art methods primarily include LDP-net Zhou et al. [2023], StyleAdv Fu et al. [2023], GNN+wave-SAN Fu et al. [2022], RDC Li et al. [2022], AFA Hu and Ma [2022], ATA Wang and Deng [2021], LRP Sun et al. [2021], FWT Tseng et al. [2019], and Fine-tuning Guo et al. [2020]. These methods can be categorized into three types: direct inference, using query samples to assist inference (marked with $\dagger$), and fine-tuning-based inference (marked with $*$).

Among these methods, direct inference (e.g., LDP-net, MatchingNet, AFA, ATA) is the most straight-forward manifestation of model generalization. It handles each test task without fine-tuning the feature embedding network, meeting practical application requirements. Using query samples to assist inference (e.g., RDC$^\dagger$, LDP-net$^\dagger$, AFA$^\dagger$, ATA$^\dagger$) is also a common experimental setting. It is noteworthy that GNN+wave-SAN$^\dagger$ and StyleAdv$^\dagger$ also leverage query samples in an unsupervised manner. The main reason is that both GNN+wave-SAN and StyleAdv use Graph Neural Network (GNN)Garcia and Bruna [2018] as a classifier. GNN treats each sample in the few-shot task as a node of the graph, and the associations between different samples as edges for reasoning, akin to label propagationLiu et al. [2019]. This approach implicitly leverages unsupervised query samples when the number of query samples in the few-shot task exceeds one. The original GNN paper Garcia and Bruna [2018] tested a single query image for each few-shot task, avoiding this issue. For a fair

Table 2: Comparison with state-of-the-art methods on 5-way 5-shot cross-domain FSL. Average classification accuracies (%) are provided. † stands for exploiting the full data of FSL task. * means that the feature embedding network needs to be fine-tuned (Ft) on each target domain tasks. The best results are in bold.

| Methods | Ft | CUB | Cars | Places | Plantae | Chest | ISIC | EuroSAT | CropDisease | Ave. |
|---|---|---|---|---|---|---|---|---|---|---|
| MatchingNet Vinyals et al. [2016] | ✗ | 51.37 | 38.99 | 63.16 | 46.53 | 22.40 | 36.74 | 64.45 | 66.39 | 48.75 |
| MAML Finn et al. [2017] | ✗ | - | - | - | - | 23.48 | 40.13 | 71.70 | 78.05 | - |
| RelationNet Sung et al. [2018] | ✗ | 56.77 | 40.46 | 64.25 | 42.71 | 24.07 | 38.60 | 65.56 | 72.86 | 50.66 |
| MetaOptNet Lee et al. [2019] | ✗ | - | - | - | - | 22.53 | 36.28 | 64.44 | 68.41 | - |
| GNN Garcia and Bruna [2018] | ✗ | 62.87 | 43.70 | 70.91 | 48.51 | 23.87 | 42.54 | 78.69 | 83.12 | 56.77 |
| FWT Tseng et al. [2019] | ✗ | 64.97 | 46.19 | 70.70 | 49.66 | 24.28 | 40.87 | 78.02 | 87.07 | 57.72 |
| LRP Sun et al. [2021] | ✗ | 64.44 | 46.20 | 74.45 | 54.46 | 24.53 | 44.14 | 77.14 | 86.15 | 58.94 |
| ATA Wang and Deng [2021] | ✗ | 66.22 | 49.14 | 75.48 | 52.69 | 24.32 | 44.91 | 83.75 | 90.59 | 60.89 |
| AFA Hu and Ma [2022] | ✗ | 68.25 | 49.28 | **76.21** | 54.26 | 25.02 | 46.01 | **85.58** | 88.06 | 61.58 |
| LDP-net Zhou et al. [2023] | ✗ | 70.39 | 52.84 | 72.90 | 58.49 | **26.67** | 48.06 | 82.01 | 89.40 | 62.60 |
| Ours | ✗ | **73.61** | **54.22** | 73.78 | **61.39** | 26.53 | **48.70** | 81.24 | **90.68** | **63.77** |
| ATA† Wang and Deng [2021] | ✗ | 65.31 | 46.95 | 72.12 | 55.08 | 23.60 | 45.83 | 79.47 | 88.15 | 59.56 |
| AFA† Hu and Ma [2022] | ✗ | 65.86 | 47.89 | 72.81 | 55.67 | 23.47 | 46.29 | 80.12 | 85.69 | 59.73 |
| RDC† Li et al. [2022] | ✗ | 63.39 | 52.75 | 72.83 | 55.30 | 25.10 | 42.10 | 79.12 | 88.03 | 59.83 |
| GNN+wave-SAN† Fu et al. [2022] | ✗ | 70.31 | 46.11 | 76.88 | 57.72 | 25.63 | 44.93 | 85.22 | 89.70 | 62.06 |
| LDP-net† Zhou et al. [2023] | ✗ | 73.34 | 53.06 | 75.47 | 59.64 | **26.88** | 48.44 | 84.05 | 91.89 | 64.10 |
| StyleAdv† Fu et al. [2023] | ✗ | 68.72 | 50.13 | **77.73** | 61.52 | 26.07 | 45.77 | **86.58** | **93.65** | 63.77 |
| Ours† | ✗ | **76.68** | **55.44** | 76.98 | **63.08** | 26.45 | **49.07** | 83.22 | 93.09 | **65.50** |
| Fine-tuning* Guo et al. [2020] | ✔ | 63.76 | 51.21 | 70.68 | 56.45 | 25.37 | 49.51 | 81.59 | 89.84 | 61.05 |
| NSAE(CE+CE)* Liang et al. [2021] | ✔ | 68.51 | 54.91 | 71.02 | 59.55 | 27.10 | 54.05 | 83.96 | 93.14 | 64.03 |
| ConFeSS* Das et al. [2021] | ✔ | - | - | - | - | 27.09 | 48.85 | 84.65 | 88.88 | - |
| ATA*† Wang and Deng [2021] | ✔ | 70.14 | 55.23 | 73.87 | 59.02 | 24.74 | 49.83 | 85.47 | 93.56 | 63.98 |
| RDC*† Li et al. [2022] | ✔ | 67.23 | 53.49 | 74.91 | 57.47 | 25.07 | 49.91 | 84.29 | 93.30 | 63.21 |

comparison, we also implement a variant that uses query samples to assist inference. Specifically, we train a classifier based on the support set to generate pseudo-labels for the query set, then filter samples from the query set based on these pseudo-labels to expand the support set, and finally retrain the classifier based on the expanded support set for the ultimate prediction on the query set.

**Results.** Tables 1 and 2 present the experimental results under 5-way 1-shot and 5-way 5-shot settings, respectively. For an easy comparison, the average performance across eight target domains is calculated as the metric. Our method achieves 46.85% (1-shot) and 63.77% (5-shot) under the direct inference setting. Compared to the second-highest method LDP-net Zhou et al. [2023], the proposed method improved by 0.51% and 1.17% on the 1-shot and 5-shot tasks, respectively. In comparison to other methods like AFA Hu and Ma [2022], ATA Wang and Deng [2021], LRP Sun et al. [2021], and FWT Tseng et al. [2019], the proposed method demonstrates greater performance advantages. Moreover, the proposed method achieves the best results on five target domains, showcasing its robust generalization ability across diverse domains. When the proposed method further utilizes query samples to assist inference, the performance is further improved. Under the same comparison, the proposed method (†) improved by 1.10% (1-shot) and 1.40% (5-shot) compared to the second-best method LDP-net†Zhou et al. [2023]. In contrast to methods based on fine-tuning (e.g., Fine-tuning*Guo et al. [2020], RDC*† Li et al. [2022]), the proposed method still achieves certain performance advantages without requiring additional fine-tuning. In summary, the proposed method has demonstrated the best cross-domain few-shot learning performance, indicating its ability to learn generalizable features in the source domain. Additionally, the method's independence from task-level embedding network fine-tuning makes it suitable for potential industrial applications.

### 3.3 Ablation study

**Comparison with baselines.** We design two baselines: the "Pre-training baseline" and the "Meta-baseline." For the "Pre-training baseline", we directly use the pre-trained model for meta-testing. The proposed method performs meta-training on the basis of pre-training. When all components are removed from the proposed method, it is equivalent to the "Meta-baseline"Chen et al. [2021]. We take the "Meta-baseline" as the baseline of the proposed method. For a fair comparison, during the meta-testing stage, these two baselines also use the same classifier as the proposed method. The

comparison results are shown in Table3. Overall, compared with these two baselines, the proposed method has achieved greater performance advantages. Specifically, compared with the "Pre-training baseline", the performance of the proposed method is improved by 2.79% (1-shot) and 3.28% (5-shot) on average. Compared with the "Meta-baseline", the performance of the proposed method is improved by 2.26% (1-shot) and 3.10% (5-shot) on average. These results show that the proposed method can improve the baselines and provide a novel meta-learning framework for CD-FSL.

Table 3: Ablation study. Average classification accuracies (%) are provided. ✔ indicates that this component is used, vice versa. The best results are in bold.

| | | CUB | | Places | | Plantae | | CropDisease | | Ave. | |
|---|---|---|---|---|---|---|---|---|---|---|---|
| Method | | 1-shot | 5-shot | 1-shot | 5-shot | 1-shot | 5-shot | 1-shot | 5-shot | 1-shot | 5-shot |
| Pretraining baseline | | 46.90 | 68.05 | 50.24 | 71.43 | 38.47 | 57.08 | 69.89 | 89.80 | 51.37 | 71.59 |
| Meta baseline | | 47.05 | 67.99 | 51.09 | 71.74 | 39.26 | 57.82 | 70.22 | 89.54 | 51.90 | 71.77 |
| Ours | | **51.55** | **73.61** | **52.06** | **73.78** | **41.55** | **61.39** | **71.47** | **90.68** | **54.16** | **74.87** |
| Alignment | Reconstruction | 1-shot | 5-shot | 1-shot | 5-shot | 1-shot | 5-shot | 1-shot | 5-shot | 1-shot | 5-shot |
| ✔ | ✗ | 50.79 | 72.65 | 51.42 | 73.22 | 41.05 | 60.93 | 70.80 | 90.11 | 53.51 | 74.22 |
| ✗ | ✔ | 50.55 | 71.39 | 51.96 | 72.60 | 41.11 | 60.22 | 70.04 | 89.44 | 53.41 | 73.41 |
| ✔ | ✔ | 51.55 | 73.61 | 52.06 | 73.78 | 41.55 | 61.39 | 71.47 | 90.68 | 54.16 | 74.87 |

**Effectiveness of the proposed frequency prior.** In this work, we propose a prediction consistency prior and a feature reconstruction prior to jointly regularize the embedding network during meta-learning. Among them, the prediction consistency prior encourages to align the predictions produced by the original query image and its each frequency component. The feature reconstruction prior aims at reconstructing the original features utilizing low-frequency and high-frequency information in the latent space, which promotes the model to learn comprehensive representations. We conduct ablation studies to illustrate the contribution of these two components. The results are shown in Table 3. As can be seen, compared to the "Meta-baseline" (meaning not using any components), the proposed method achieves average gains of 1.61% (1-shot) and 2.45% (5-shot). The above experimental results show the proposed prediction consistency prior is effective. We can draw similar conclusions for feature reconstruction prior. In addition, the proposed method improves by nearly 0.65% under both 1-shot and 5-shot tasks when the feature reconstruction module is added. In particular, for the CUB dataset, the proposed method can achieve nearly 1% improvement on the 5-shot task. This indicates that the proposed feature reconstruction prior is beneficial to the entire method.

## 3.4 Visualization

**Feature highlight.** we adhere to established practices Zhou et al. [2023], utilizing the model trained on the source domain to extract features from target domain images. Subsequently, these features serve as attention scores to activate the original images. The results are presented in Fig.2. Overall, our proposed method exhibits the capability to capture more nuanced representations compared to the baseline, a critical aspect for effective cross-domain generalization. As an illustrative example, consider image (d) in Fig.2. The baseline tends to concentrate solely on the neck of the bird, neglecting the broader characteristics of its entire shape. In contrast, our method not only hones in on local texture details, such as the head, wings, and claws, but also encapsulates the entirety of the contour shape. This underscores the capacity of our method to learn comprehensive features, avoiding undue emphasis on local textures alone.

**Domain gap.** The t-SNE Van der Maaten and Hinton [2008] visual results are shown in Fig.3 (a-b). The blue cluster represents the source domain distribution, while the other four colors denote distinct target domain distributions. Notably, the baseline exhibits a substantial gap between target domains and the source domain. In contrast, our method effectively mitigates this domain gap. In addition, we conduct a quantitative assessment of the distribution distance between different target domains and the source domain. Specifically, we compute the first-order statistics based on the sampled samples in each domain, treating it as the statistical characteristic of that domain. Subsequently, we measure the Euclidean distance between the first-order statistics of different target domains and the source domain. The resulting quantitative metrics, comparing the proposed method and the baseline, are visualized in Fig.3 (c). Evidently, the proposed method exhibits a smaller distribution distance between the source domain and the target domains, with particularly notable improvements in the medical domain (ISIC)

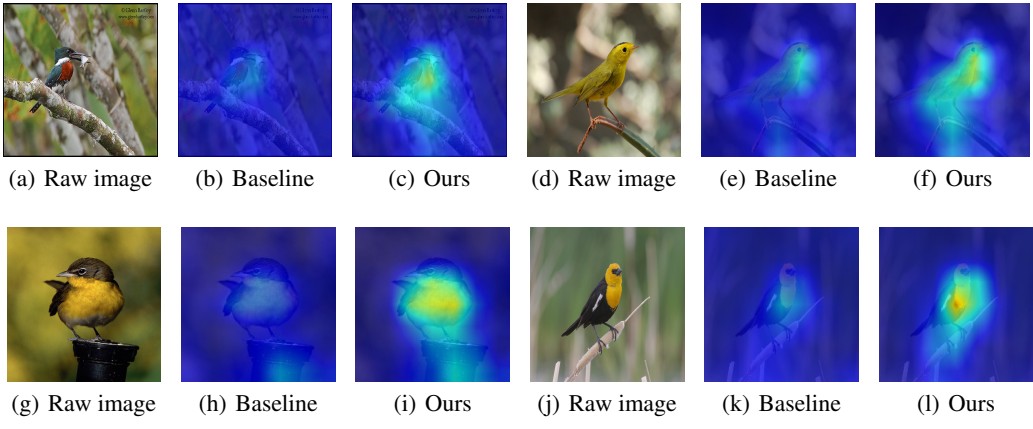

| (a) Raw image | (b) Baseline | (c) Ours | (d) Raw image | (e) Baseline | (f) Ours |
|---|---|---|---|---|---|
| (g) Raw image | (h) Baseline | (i) Ours | (j) Raw image | (k) Baseline | (l) Ours |

Figure 2: Feature visualization for Baseline and the proposed method.

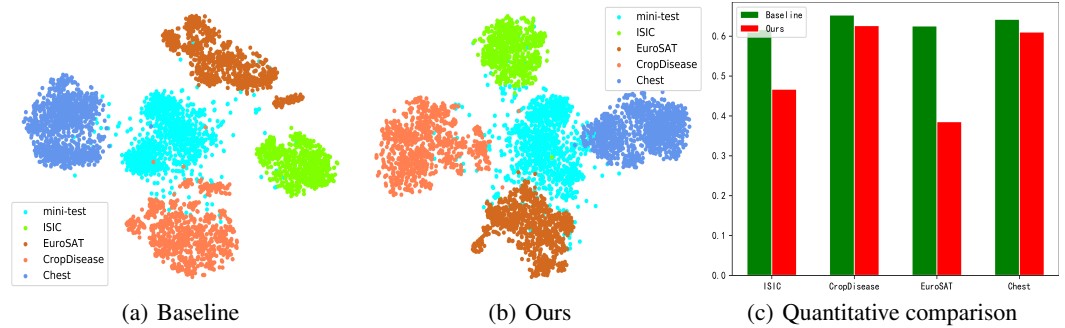

(a) Baseline        (b) Ours        (c) Quantitative comparison

Figure 3: Visual observations in domain gap.

and the remote sensing domain (EuroSAT). This underscores the capability of our method to learn robust representations in the source domain, effectively mitigating domain shifts.

## 4 Conclusions

In this work, we propose an insightful meta-learning framework inspired by the cross-domain invariant frequency priors. Furthermore, we present a prediction consistency prior and a feature reconstruction prior to jointly regularize meta-learning on source domain, enabling learning cross-domain transferable features. This simple yet effective work achieves state-of-the-art experimental results as well as excellent inference efficiency.

**Limitations.** The limitation of the proposed method lies on its robustness in some extremely challenging cross-domain tasks. For example, on the Chest dataset, the proposed method fails to outperforms all competitors. This indicates that the fixed image decomposition (e.g., Fast Fourier Transform or Wavelet Transform) strategy may be not the optimal solution for all unknown cases in terms of exploit frequency priors. In the future, we will attempt to exploit the learnable image decomposition strategy. In addition, the proposed method requires to decompose the query image before being fed into the network. While the Fast Fourier Transform for signal decomposition is efficient, this does introduce a certain additional training time overhead. Notably, this work does not contain negative social impact.

**Acknowledge.** This work was supported in part by the National Natural Science Foundation of China under Grand 62372379, Grant 62071387, Grant 62472350, Grant 62472359 and Grant 62101454; in part by the National Key R&D Program of China under Grand 2022ZD0118700; in part by the Xi'an's Key Industrial Chain Core Technology Breakthrough Project: AI Core Technology Breakthrough under Grand 23ZDCYJSGG0003-2023; in part by Innovation Foundation for Doctor Dissertation of Northwestern Polytechnical University under Grant CX2024017; in part by National Key Laboratory of Science and Technology on Space-Born Intelligent Information Processing fundation under Grant TJ-04-23-04.

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

# A Related Work

**Few-shot learning.** Early investigations into model generalization with limited data primarily focused on few-shot learning (FSL), giving rise to a series of seminal meta-learning methods Vinyals et al. [2016], Snell et al. [2017], Huang et al. [2022], Zhang and Huang [2022], Finn et al. [2017], Lee et al. [2019], Rusu et al. [2019], Zhmoginov et al. [2022], Baik et al. [2020], Zhang et al. [2020] aimed at addressing FSL challenges. In the realm of architecture, a typical meta-learning model comprises a task-agnostic meta-learner and a task-specific base-learner. Notably, ProtoNet Snell et al. [2017] and MatchingNet Vinyals et al. [2016] construct the base-learner using a non-parametric distance measure, while Meta-opt Lee et al. [2019] and DeepEMD Zhang et al. [2020] employ a differentiable linear classifier for this purpose. On the optimization front, meta-learning methods Vinyals et al. [2016], Snell et al. [2017], Huang et al. [2022], Zhang and Huang [2022], Finn et al. [2017], Lee et al. [2019], Rusu et al. [2019], Zhmoginov et al. [2022], Baik et al. [2020] typically employ a two-stage optimization strategy. Initially, they optimize the base-learner based on a limited set of labeled data, followed by optimizing the meta-learner to minimize empirical risk on unlabeled data. Despite the progress made by these methods, they exhibit limited generalization capabilities when confronted with cross-domain tasks Guo et al. [2020].

**Cross-domain few-shot learning.** Several recent advancements Zhou et al. [2023], Fu et al. [2022, 2023], Li et al. [2022], Wang and Deng [2021], Hu and Ma [2022], Guo et al. [2020], Liang et al. [2021], Das et al. [2021], Li et al. [2021], Fu et al. [2021], Sun et al. [2021] have concentrated on few-shot learning (FSL) within cross-domain scenarios, where the target and source domains differ not only in category but also in domain distribution. Cross-domain few-shot learning (CD-FSL) presents significant challenges and opportunities, given that the distribution of target tasks in practical applications often deviates from that of the source domain. Moreover, acquiring data for extreme target domains, such as medical images Tschandl et al. [2018], Wang et al. [2017], or annotating remote sensing scene images Helber et al. [2019], is frequently arduous.

Zhou et al. Zhou et al. [2023] consider that local features are robust to cross-domain tasks and propose an improved ProtoNet Snell et al. [2017] to help the model focus on local regions of the image to avoid the simplicity bias. Fu et al. Fu et al. [2022] employ style augmentation during model training as a strategy to mitigate the detrimental effects on generalization caused by style variations in the target domain. Building upon the ideas presented in their earlier work Fu et al. [2022], Fu et al. Fu et al. [2023] extend their approach by incorporating adversarial training. This additional step aims to assist the model in adapting to domain shifts, enhancing its robustness across different domains. Li et al. Li et al. [2022] utilize target task information to perform distance calibration on the embedding of the source domain model to promote generalization. Wang et al. Wang and Deng [2021] and Hu et al. Hu and Ma [2022] design adversarial training methods to simulate domain changes from the task-level and feature-level respectively. In this way, the model can obtain domain-invariant representations. Guo et al. Guo et al. [2020] propose a cross-domain fine-tuning baseline, which fine-tunes the feature extraction model for each target domain task. Liang et al. Liang et al. [2021] further design a self-supervised reconstruction loss to fine-tune the model, which can help the model learn more comprehensive representations. Although fine-tuning based methods Guo et al. [2020], Liang et al. [2021] achieve good performance, they require a large number of iterative training for each target domain task. This paradigm brings additional computational and storage overhead. In contrast, our proposed method prioritizes enabling the model to learn cross-domain generalization knowledge during the training phase, allowing for generalization across various target domains without relying on fine-tuning. Additionally, orthogonal to the aforementioned methods Zhou et al. [2023], Fu et al. [2023], Li et al. [2022], Wang and Deng [2021], Hu and Ma [2022], Guo et al. [2020], Liang et al. [2021], Das et al. [2021], Li et al. [2021], Fu et al. [2021], Sun et al. [2021], our approach centers on utilizing cross-domain invariant frequency priors to alleviate the over-fitting problems of classic meta-learning, facilitating the acquisition of cross-domain transferable features.

**Different image priors.** In the realm of classical image processing, researchers historically devised effective pattern extractors based on diverse image priors such as texture Guo et al. [2010], Pathak and Barooah [2013] and shape Vincent et al. [2009], Ding and Goshtasby [2001]. However, with the advent of deep learning, Convolutional Neural Networks (ConvNets) have demonstrated remarkable proficiency in capturing texture features but have often struggled to encapsulate critical shape priors Geirhos et al. [2018], Hermann et al. [2020], Ringer et al. [2019], Jain et al. [2022]. To

address this limitation, recent works Stojanov et al. [2021], Padmanabhan et al. [2023], Heo et al. [2023] have employed shape priors to mitigate texture bias and enhance model generalization. For instance, Jain et al.Jain et al. [2022] advocate for incorporating both texture and shape priors to bolster model generalization and mitigate spurious correlations. In Few-Shot Learning (FSL), Stojanov et al.Stojanov et al. [2021] employ point clouds to explicitly derive shape priors, subsequently minimizing the distance between point cloud embeddings and image embeddings to alleviate texture bias. Similarly, Padmanabhan et al.Padmanabhan et al. [2023] utilize the Sobel operatorVincent et al. [2009] to extract object shape priors and integrate shape-aware knowledge to enhance model generalization.

On a different front, certain studies Yin et al. [2019], Fu et al. [2022], Chen and Wang [2021], Zhao et al. [2022], Cheng et al. [2023] have underscored the positive impact of frequency priors on model generalization. For instance, Yin et al.Yin et al. [2019] observed distinct robustness levels of high-frequency and low-frequency components to noise, inspiring researchers to employ frequency domain data augmentation for enhanced model generalizationZhao et al. [2022], Cheng et al. [2023]. In FSL, Chen et al.Chen and Wang [2021] concatenate frequency domain features with original image features to obtain comprehensive representations. Fu et al.Fu et al. [2022] propose exchanging high-frequency and low-frequency components between different images for image style augmentation. Cheng et al. Cheng et al. [2023] leverage gradient information to identify areas with higher activation levels in frequency domain images.

In contrast to methods relying on texture or shape priors, the proposed method starts from the principles of image transformation theory, focusing on cross-domain invariant frequency priors to enhance the robustness of model. Additionally, our approach compels the model to simultaneously attend to both high-frequency structure and low-frequency content. This strategy enables our method to strike a balance between the contributions of texture and shape to model generalization. Furthermore, unlike other methods grounded in frequency priors, our work's primary innovation lies in the seamless integration of low-frequency and high-frequency features within a meta-learning framework. This integration provides an elegant solution to the persistent challenges of cross-domain few-shot learning, allowing for the independent learning of features from these distinct segments, each excelling in capturing unique aspects of visual information.

Table 4: Ablation on different priors. Average classification accuracies (%) are provided. The best results are in bold.

| Method | CUB | | Places | | Plantae | | CropDisease | | Ave. | |
|---|---|---|---|---|---|---|---|---|---|---|
| | 1-shot | 5-shot | 1-shot | 5-shot | 1-shot | 5-shot | 1-shot | 5-shot | 1-shot | 5-shot |
| Texture | 48.68 | 70.31 | 51.12 | 72.66 | 39.41 | 58.72 | 70.09 | 89.80 | 52.32 | 72.87 |
| Shape | 45.02 | 67.14 | 49.95 | 71.15 | 36.27 | 56.97 | 65.81 | 87.18 | 49.26 | 70.61 |
| Texture+Shape | 48.08 | 70.00 | 51.27 | 72.60 | 38.49 | 58.66 | 69.19 | 89.19 | 51.75 | 72.61 |
| Low frequency | 50.05 | 71.58 | 51.31 | 72.78 | 40.72 | 60.28 | 70.05 | 89.90 | 53.03 | 73.63 |
| High frequency | 49.89 | 71.61 | 51.14 | 72.84 | 40.65 | 60.49 | 70.04 | 89.95 | 52.93 | 73.72 |
| Ours | **51.55** | **73.61** | **52.06** | **73.78** | **41.55** | **61.39** | **71.47** | **90.68** | **54.16** | **74.87** |

## B   Further analysis

**Why exploit frequency prior?**   As discussed in Sec. A, previous works Stojanov et al. [2021], Padmanabhan et al. [2023], Heo et al. [2023], Yin et al. [2019], Chen and Wang [2021], Zhao et al. [2022], Cheng et al. [2023] have explored the integration of various priors, including texture, shape, and frequency, among others. To highlight the advantages of our proposed method, we conducted comparative experiments with different variants. To ensure a fair comparison, we replaced different priors in our framework while maintaining other settings constant. Specifically, we adopted the approach of Jain et al. [2022] to model the texture prior. For the shape prior, similar to Stojanov et al. [2021], Padmanabhan et al. [2023], we employed the Canny operator Ding and Goshtasby [2001] to extract the shape prior. For high-frequency or low-frequency priors, we retained only the high-frequency branch or the low-frequency branch in our framework. The results are presented in Table 4. Overall, our method outperforms the variants with different priors. The reasons behind this superiority are as follows. Firstly, the texture prior compels the model to excessively focus on local discriminative regions, leading to texture bias and impairing generalization. Secondly, the

shape prior directs the model's attention to global shapes, which may cause the model to exhibit shape bias and overlook semantic information. Thirdly, compared to texture or shape priors, the frequency prior provides more original information, expanding the model's search space and enabling a higher generalization upper bound. Fourthly, our method couples high-frequency and low-frequency information within a unified framework, presenting an elegant solution to the persistent challenges of cross-domain few-shot learning. This approach allows for the simultaneous consideration of both types of information, harnessing their complementary aspects for improved generalization.

**Why can our work alleviate overfitting?**    In this study, we subscribe the limited generalization capacity of exiting methods in cross-domain few-shot learning (CD-FSL) to their over-fitting onto source domain. Since during the meta-training procedure, only tasks randomly sampled from source domain are utilized for model update, when the target domain shows obvious distribution discrepancy from the source domain, these existing methods are prone to over-fitting, in other words, fail to generalize well in the new target domain. Why can our work alleviate overfitting? To solve over-fitting, a direct solution is to introduce appropriate prior (e.g., regularization) during training on source domain. Inspired by this, we attempt to comprehensively exploit the cross-domain transferable image frequency prior that each image can be decomposed into complementary low-frequency content details and high-frequency robust structural characteristics. Following this idea, we first utilizes Fast Fourier Transform to explicitly decouple the high-frequency and low-frequency components of the image. Then, we feed each component and the query image into a three-branch feature embedding network for category prediction. More importantly, we further establish a feature reconstruction prior and a prediction consistency prior to collectively guiding the network's meta-learning process. The feature reconstruction prior requires to reconstruct the feature of original image through fusing the features of both decomposed frequency parts using a deep projection network, while the prediction consistency prior aims to minimize the separate Kullback-Leibler divergence between the prediction scores produced by the original query image and its each frequency component. By doing these, both priors encourage to exploit a deep feature space where no matter full-frequency band (original image), high-frequency component or low-frequency component can lead to the unique and correct classification prediction, i.e., the idea semantic feature space which is transferable cross-domain. Therefore, the proposed method is able to mitigate the over-fitting problems in CD-FSL. Our state-of-the-art performance on eight benchmark datasets as well as the ablation study also support this conclusion.

**Whether this work is equivalent to data augmentation or self-supervised learning?**    In this work, our core idea is to utilize cross-domain invariant frequency priors to alleviate the over-fitting problem of classical meta-learning in cross-domain few-shot learning tasks. To this end, we propose two key components: the Image Decomposition Module (IDM) and the Prior Regularization Meta-Network (PRM-Net). Among them, IDM aims to use Fast Fourier Transform (FFT) to explicitly decompose each image from few-shot task into its low- and high-frequency components Nussbaumer and Nussbaumer [1982]. PRM-Net is a key component responsible for introducing a prediction consistency prior and a feature reconstruction prior. It is important to note that our method is fundamentally different from simple data augmentation methods and self-supervised learning methods. The insight behind our method is divide and conquer, that is, explicit decomposition and implicit coupling. First of all, the IDM is to explicitly obtain the frequency priors of the image rather than to simply perform data augmentation. IDM provides frequency prior for the subsequent PRM-Net, and its role is "divide". In addition, PRM-Net designs regular terms with the help of frequency priors, rather than simple self-supervised learning, and its role is "conquer". More importantly, this divide-and-conquer strategy enables IDM and PRM-Net to collaborate to provide a powerful meta-learning framework, aiming to enhance cross-domain generalization by explicitly considering image decomposition and introducing effective regularization during the meta-learning process. We design some experiments to compare the proposed method with other data augmentation methods and self-supervised learning methods. For the data augmentation method, we use random rotation to augment the image, and design the angular self-supervised loss as the regularization term Gidaris et al. [2018]. For self-supervised learning methods, we choose the most representative SimCLR Chen et al. [2020] and BYOL Grill et al. [2020]. All compared methods keep the same backbone network and training data as the proposed method. The experimental results are shown in Table 5. Overall, the proposed method can achieve better results compared to using simple data augmentation and self-supervised learning methods.

Table 5: Comparison with other data augmentation methods and self-supervised learning methods. Average classification accuracies (%) are provided. The best results are in bold.

| | CUB | | Places | | Plantae | | CropDisease | | Ave. | |
|---|---|---|---|---|---|---|---|---|---|---|
| Method | 1-shot | 5-shot | 1-shot | 5-shot | 1-shot | 5-shot | 1-shot | 5-shot | 1-shot | 5-shot |
| Rotation augmentation | 49.04 | 71.17 | 50.07 | 72.79 | 39.90 | 59.33 | 69.52 | 90.59 | 52.13 | 73.47 |
| SimCLR | 46.40 | 69.08 | 50.78 | 72.86 | 39.77 | 59.65 | **72.50** | **91.56** | 52.36 | 73.28 |
| BYOL | 47.96 | 70.13 | 49.48 | 71.88 | 40.38 | 59.73 | 71.91 | 91.15 | 52.43 | 73.22 |
| Ours | **51.55** | **73.61** | **52.06** | **73.78** | **41.55** | **61.39** | 71.47 | 90.68 | **54.16** | **74.87** |

**Whether it is applicable to any image decomposition method?** To answer this question, we conducted comparative experiments using FFT-based Nussbaumer and Nussbaumer [1982] decomposition and Wavelet-based Zhang and Zhang [2019] decomposition. The results are presented in Table 6. In comparison to the baseline, our method consistently demonstrates significant performance advantages regardless of the decomposition method employed. This indicates the scalability and effectiveness of our method across different decomposition techniques. Additionally, the model trained with Wavelet decomposition exhibits further performance improvement on the Places dataset.

Table 6: Comparison with different image decomposition methods. Average classification accuracies (%) are provided. The best results are in bold.

| | CUB | | Places | |
|---|---|---|---|---|
| Method | 1-shot | 5-shot | 1-shot | 5-shot |
| Baseline | 47.05 | 67.99 | 51.09 | 71.74 |
| Haar-wavelet | 49.12 | 71.12 | **52.63** | **73.92** |
| FFT | **51.55** | **73.61** | 52.06 | 73.78 |

| | Plantae | | CropDisease | |
|---|---|---|---|---|
| Method | 1-shot | 5-shot | 1-shot | 5-shot |
| Baseline | 39.26 | 57.82 | 70.22 | 89.54 |
| Haar-wavelet | 40.56 | 60.46 | 70.82 | 90.45 |
| FFT | **41.55** | **61.39** | **71.47** | **90.68** |

**Does performance benefit from additional parameters?** First of all, we clarify that our method does not introduce additional learnable parameters and the parameter amount is same as our baseline. This is because only the parameters in the main branch (e.g., query image branch) are learnable, and the parameters of the high-frequency branch and low-frequency branch are updated through the exponential moving average of the main branch parameters. Moreover, after training, we only keep the main branch for prediction in the test phase, since the prediction consistency prior have forced these three branches to produce the same prediction results when the training procedure converged. Therefore, our method does not introduce any additional inference costs compared with our baseline. We also supplemented experiments to answer whether the performance gain comes from additional parameters. Specifically, we triple the learnable parameters of the baseline method and then compare it with our method. As shown in Table 7, our method still outperforms the baseline when the parameters of the baseline are increased three times.

**Efficiency.** As mentioned previously, the proposed method focuses on obtaining a generalizable model through meta-learning without relying on fine-tuning the embedding network on the target domain. This makes the proposed method very practical and efficient in handling target domain tasks. To verify the efficiency of the proposed method, we chose a classic fine-tuning based method Guo et al. [2020] for comparison. For the fine-tuning based method, we follow its original settings for experiments. We report the average classification accuracy across different target domains as well as the required inference time for each target domain task. Our experimental platform is a single

Table 7: Compared three times baseline with ours. Average classification accuracies (%) are provided. The best results are in bold.

| Method | CUB | | Places | | Plantae | | CropDisease | | Ave. | |
|---|---|---|---|---|---|---|---|---|---|---|
| | 1-shot | 5-shot | 1-shot | 5-shot | 1-shot | 5-shot | 1-shot | 5-shot | 1-shot | 5-shot |
| Baseline | 47.05 | 67.99 | 51.09 | 71.74 | 39.26 | 57.82 | 70.22 | 89.54 | 51.90 | 71.77 |
| Baseline@3x | 48.16 | 69.42 | 51.51 | 72.20 | 39.25 | 58.14 | 70.83 | 90.21 | 52.43 | 72.49 |
| Ours | **51.55** | **73.61** | **52.06** | **73.78** | **41.55** | **61.39** | 71.47 | 90.68 | **54.16** | **74.87** |

3090 GPU. The results are shown in Table 8. It can be seen that the proposed method only requires about 0.06 seconds to effectively handle a few-shot task in the target domain. Compared with the fine-tuning based method, the proposed method is nearly 100 times faster on the 5-shot task. At the same time, the proposed method also has great advantages in performance. The above experiments show that the proposed method is efficient and has the potential for practical applications.

Table 8: Comparison of efficiency between the proposed method and fine-tuning method. The average inference time on each task is reported. The best results are in bold.

| Method | 1-shot | | 5-shot | |
|---|---|---|---|---|
| | Efficiency $\downarrow$ | Accuracy $\uparrow$ | Efficiency $\downarrow$ | Accuracy $\uparrow$ |
| Fine-tuning Guo et al. [2020] | 2.21 | 45.54 | 7.67 | 61.05 |
| Ours | **0.06** | **46.85** | **0.07** | **63.77** |

**Hyper-parameters validation.** The hyper-parameters of the proposed method encompass the momentum parameters $m_1$ and $m_2$, which are integral to the Exponential Moving Average (EMA) update strategy. In our approach, we employ the loss for calculating gradients and subsequently updating the network parameters of the main branch through back-propagation. Conversely, for the parameters in the low-frequency and high-frequency branches, we update them using EMA. To validate the advantages of EMA, we conducted a parameter sharing experiment for comparison, denoted as "none" in Table 9. The results indicate a substantial performance decrease when using parameter sharing as opposed to EMA, underscoring the necessity of employing EMA. Additionally, we explored different values for the momentum parameters and observed that the model's performance is not highly sensitive to the specific values of $m_1$ and $m_2$. We have set $m_1$ to 0.997 and $m_2$ to 0.999 based on these findings.

**Additional visualization.** More feature highlight results are provided in Fig.4 and Fig.5. In summary, the baseline model tends to focus narrowly on specific local regions of the object. Conversely, our method exhibits a more extensive focus on the object, indicating an ability to capture a more comprehensive semantic understanding and, consequently, achieve superior generalization.

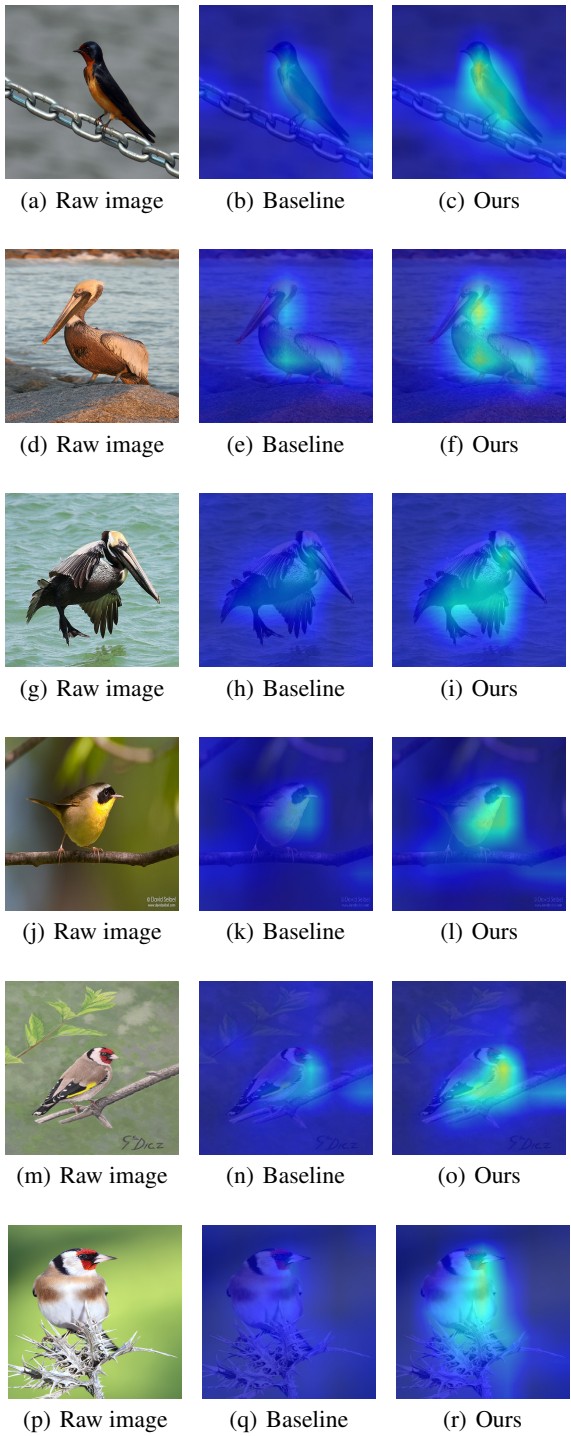

(a) Raw image     (b) Baseline     (c) Ours

(d) Raw image     (e) Baseline     (f) Ours

(g) Raw image     (h) Baseline     (i) Ours

(j) Raw image     (k) Baseline     (l) Ours

(m) Raw image     (n) Baseline     (o) Ours

(p) Raw image     (q) Baseline     (r) Ours

Figure 4: Feature visualization for baseline and the proposed method.

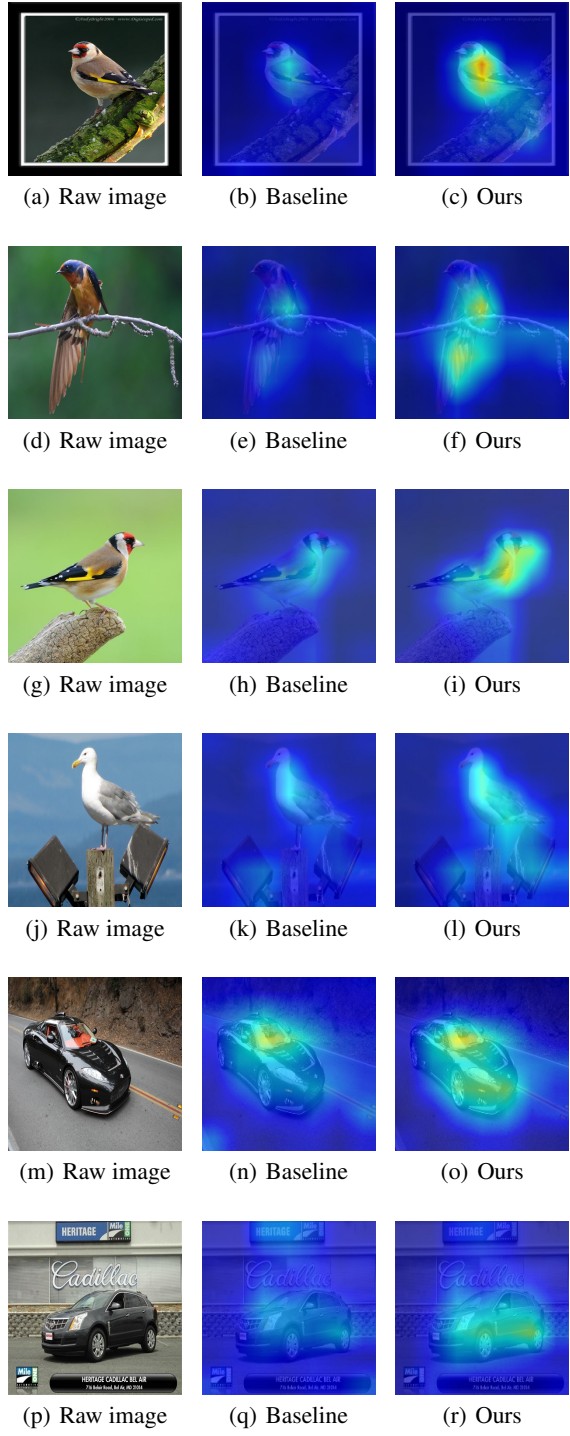

Figure 5: Feature visualization for baseline and the proposed method.

Table 9: Classification accuracy w.r.t values of momentum. Average classification accuracies (%) are provided. The best results are in bold.

| | CUB | | Places | |
|---|---|---|---|---|
| Value | 1-shot | 5-shot | 1-shot | 5-shot |
| none | 50.76 | 72.56 | 51.83 | 72.76 |
| $m_1$=0.9999, $m_2$=0.9995 | 51.28 | 73.33 | 52.10 | 73.75 |
| $m_1$=0.9999, $m_2$=0.9997 | 51.44 | 73.44 | **52.14** | **73.84** |
| $m_1$=0.9997, $m_2$=0.9999 | **51.55** | **73.61** | 52.06 | 73.78 |
| $m_1$=0.9995, $m_2$=0.9999 | 51.52 | 73.54 | 52.00 | 73.72 |

| | Plantae | | CropDisease | |
|---|---|---|---|---|
| Value | 1-shot | 5-shot | 1-shot | 5-shot |
| none | 40.73 | 60.46 | 69.47 | 89.56 |
| $m_1$=0.9999, $m_2$=0.9995 | 41.37 | 61.30 | 71.11 | 90.38 |
| $m_1$=0.9999, $m_2$=0.9997 | 41.44 | 61.10 | 71.32 | 90.45 |
| $m_1$=0.9997, $m_2$=0.9999 | **41.55** | **61.39** | **71.47** | **90.68** |
| $m_1$=0.9995, $m_2$=0.9999 | 41.25 | 61.23 | 71.23 | 90.65 |

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
