# OpenReview forum: "Meta-Exploiting Frequency Prior for Cross-Domain Few-Shot Learning"
_NeurIPS.cc/2024/Conference — NeurIPS 2024 poster_

### Official Review · Reviewer_HqaU · 2024-06-21

**Soundness:** 3
**Presentation:** 3
**Contribution:** 2
**Rating:** 6
**Confidence:** 4

**Summary:**

This paper introduces a novel framework for cross-domain few-shot learning. The input images are first decomposed into low-frequency content and high-frequency structure using FFT. Then, the PRM-Net includes three branches, low-frequency, high-frequency, and main branch. The PRM-Net includes two priors to regularize the feature embedding network. The approach shows state-of-the-art results on multiple benchmarks.

**Strengths:**

1. The use of frequency decomposition to address CD-FSL task is easy to understand and implement.
2. The method is thoroughly evaluated on multiple benchmarks.
3. The method achieves state-of-the-art results on multiple benchmarks.
4. The paper provides a clear and detailed description of each component of the method.

**Weaknesses:**

1. The method avoids additional inference costs, however, the decomposition and additional branches (high-low frequency branches) could introduce significant computational overhead. Moreover, the improvement by incorporating the reconstruction prior seems marginal according to Table 3.
2. Lack of proper citation for the datasets mentioned in section 3.1, e.g., CropDisease, CUB, ISIC.
3. The reliance on fixed decomposition strategies like FFT might not be optimal for some scenarios, e.g., the foreground objects have similar colors to the background, or the background is complex. The whole model cannot be trained end-to-end due to FFT.

**Questions:**

1. How do you see the value of research in the (cross-domain) few-shot learning while the foundation models can perform well in zero-shot tasks?
2. Is it possible to combine this approach with fine-tuning strategies to further enhance performance in certain domains?

---

> ### Author Rebuttal · Authors · 2024-08-06
>
> ## To Reviewer HqaU :
>
> ### Weaknesses (1)
>
> ### Response 1 : About computational overhead.
>
> 1) Thanks for the comments. The proposed method indeed introduces additional computational cost during the training phase. We will further clarify this weakness in the manuscript.
>
> 2) We present the computational costs of our method during the training and inference phases in Table 2 and  3 (Due to character limitations, please see  "To Reviewer SF8W".), respectively.  As shown, the backbone and the feature reconstruction network in our method are lightweight, which to some extent mitigates the computational overhead of the high and low-frequency branches. In addition, a portion of the computational overhead of our method comes from the FFT. In future work, we will explore the use of efficient, learnable image decomposition techniques, employing differentiable kernels to capture information from different frequency bands of the images.
>
> 3) During the inference, the proposed method does not introduce additional computational overhead, resulting in efficient inference and good generalization.
>
> ### Response 2 : About the reconstruction prior.
>
> We have analyzed the reconstruction prior in the manuscript (please see "Effectiveness of the proposed frequency prior" in 4.3. Ablation study.) and performed experiments (please see Table 3 in the manuscript). We copy the results from the manuscript here, as shown in Table 12. It can be seen that compared with the baseline, the proposed reconstruction prior can achieve better performance. This demonstrates the contribution of the proposed reconstruction prior.
>
> **Table 12 : Verifying  the alignment and reconstruction prior under 5-way 1-shot (5-way 5-shot ) setting.**
> |　Method　|　CUB　|　Places　|　Plantae　|　CropDisease　|　Ave.　|
> |---|---|---|---|---|---|
> |　Baseline　|　47.05 (67.99) |　51.09 (71.74) 　|　39.26 (57.82) 　| 　 70.22 (89.54) 　|　 51.90 (71.77) 　|
> |　Ours just alignment 　|　50.79 (72.65) |　51.42 (73.22) 　|　41.05 (60.93) 　| 　70.80 (90.11) 　|　53.51 ( 74.22) 　|
> |　Ours just reconstruction　|　50.55 (71.39)　| 　51.96 (72.60)　| 　41.11 (60.22)　| 　70.04 (89.44)　| 　53.41 (73.41)　|
> |　Ours (alignment + reconstruction) |　51.55 (73.61 ) 　|　52.06 (73.78)|　41.55 (61.39)　| 　71.47 (90.68)|　54.16 (74.87)　|
>
> ### Weaknesses (2)
>
> ### Response :
>
> Thanks for the suggestion. We will supplement the references for CropDisease, CUB, and ISIC in the manuscript.
> ### Weaknesses (3)
>
> ### Response :
>
> Thanks for the comments. We will further clarify this weakness in the manuscript. To alleviate this problem, we will explore the use of learnable image decomposition methods as an alternative to FFT. Essentially, FFT uses fixed kernels for signal decomposition, making it difficult to achieve data-adaptive decomposition. We are considering designing differentiable kernels to separately extract high-frequency and low-frequency information from images. Additionally, we consider constructing a model that generates kernel parameters conditioned on the original image for data-adaptive image decomposition. We will validate it in future work.
>
> ### Questions (1)
>
> ### Response :
>
> We think that it is valuable to study cross-domain few-shot learning (CD-FSL) in the era of foundation models. The reasons are as follows.
>
> 1) Adapting foundation models to CD-FSL has practical value. The core elements of CD-FSL are feature initialization and rapid adaptation. Early research has concentrated on using meta-learning methods to learn a good feature initialization model in the source domain, and then quickly adapting the model to the target task with a small number of labeled samples.The foundation models essentially accomplished the first step of FSL. In future research, effectively adapting large models to cross-domain few-shot tasks is valuable.
>
> 2) Exploring the adaptation of foundation models to extreme CD-FSL holds significant research value.
> Previous work [1] explored applying pre-trained foundation models to extreme cross-domain few-shot tasks. For comparison, we implemented our method under the same settings, and the results are shown in Table 13. As can be seen, even with the assistance of pre-trained foundation models, these methods still perform poorly in extreme cross-domain scenarios (e.g., Chest, ISIC, etc.).
> Therefore, exploring the adaptability of foundational models to extreme CD-FSL tasks is valuable.
>
> **Table 13 : Performance using the foundation model as feature initialization.**
> |Method|Backbone|Setting|CUB|Cars|Places|Plantae|Chest|ISIC|EuroSAT|CropDisease|Ave.|
> |---|---|---|---|---|---|---|---|---|---|---|---|
> |　P>M>F[2]　|　ViT-S (DINO pretrain)　|　1-shot　|　78.13　|　37.24　|　71.11　|　53.60　|　21.73　|　30.36　|　70.74　|　80.79　|　55.46　|
> |　**Ours**　|　ViT-S (DINO pretrain)　|　1-shot　|　85.07　|　42.82　|　73.71　|　56.6　|　23.32　|　35.13　|　72.03　|　81.82　|　58.81　|
> |　P>M>F[2]　|　ViT-S (DINO pretrain)　|　5-shot　|　-　|　-　|　-　|　-　|　27.27　|　50.12　|　85.98　|　92.96　|　-　|
> |　**Ours**　|　ViT-S (DINO pretrain)　|　5-shot　|　97.5　|　74.6　|　89.24　|　81.63　|　26.31　|　51.72　|　89.95　|　95.93　|　75.86　|
>
> [1] Pushing the Limits of Simple Pipelines for Few-Shot Learning: External Data and Fine-Tuning Make a Difference, CVPR 2022.
>
> ### Questions (2)
> ### Response :
>
> Following the suggestion, we have fine-tuned our method on target domain. Following the common fine-tuning strategies, for each few-shot task in the target domain, we fine-tune the model using the support set, and then test it on the query set. Due to time constraints, we have conducted experiments on the EuroSAT and ISIC target datasets as examples, as shown in Table 14. It can be seen that, through fine-tuning, the performance of the proposed method can be further improved. For example, on the EuroSAT dataset, fine-tuning the proposed method improved performance by 1.55%.
>
> **Table 14 : Performance when fine-tuning our method under 5-way 5-shot setting.**
> |　Method　|　EuroSAT　|　ISIC 　|
> |---|---|---|
> |　Ours 　|　81.24　|　48.70　|
> |　Ours+fine-tuning　|　 82.79 |　49.50　|

---

### Official Review · Reviewer_mWuv · 2024-07-09

**Soundness:** 4
**Presentation:** 4
**Contribution:** 3
**Rating:** 7
**Confidence:** 5

**Summary:**

The paper introduces an innovative framework that leverages the concept of frequency priors for cross-domain few-shot learning. The novel idea of decomposing images into high and low-frequency components and integrating these into the meta-learning process is a creative advancement in the field.
Built upon established image transformation theories such as the Fourier Transform, the proposed method is grounded in a solid theoretical foundation, which enhances its credibility and applicability across various domains. The empirical validation through extensive experiments on multiple cross-domain benchmarks further substantiates the effectiveness of the framework, showcasing its superiority over state-of-the-art methods. And the experiment, along with the supplementary materials, has been very comprehensive, addressing most of my questions.

**Strengths:**

The paper is commendably structured, with a clear and concise presentation of ideas, and it exhibits a high degree of reproducibility, further enhanced by the open sourcing of some of the code. The abstract and introduction effectively encapsulate the motivation and key contributions of the work, providing readers with a quick yet comprehensive understanding. The methodology is explained in a step-by-step fashion, making it accessible to readers who may not be intimately familiar with meta-learning or frequency domain analysis, thereby demonstrating the paper's strength in both clarity and comprehensiveness.

**Weaknesses:**

(1) While the paper highlights the efficiency advantages of the proposed method, it falls short of offering a detailed analysis of computational complexity. A critical evaluation extending beyond inference times to include the total floating-point operations (FLOPs) during both training and inference phases is essential. Understanding the complete computational costs provides insights into the method's practicality, scalability, and suitability for deployment across varying computational environments.
(2) The paper could be improved by including more extensive ablation studies on the choice of backbone network and loss functions. The current presentation lacks a thorough exploration of why specific architectural decisions and loss formulations were made, which is crucial for understanding the contribution of these choices to the overall performance.
(2) Although the paper demonstrates strong empirical results, there is a need for a more profound theoretical analysis underpinning the effectiveness of frequency priors in cross-domain generalization. A deeper theoretical understanding would bolster the paper's claims and offer insights into the broader applicability of the proposed method.

**Questions:**

(1) Is there a hyperparameter used during the FFT process to determine the high and low-frequency information? If so, how was the threshold for decoupling these frequencies determined, and how might it influence the experimental results?
(2) How does the model perform over a longer period of training, and are there any signs of degradation or improvement in performance? Can the loss optimization curve and loss landscape be provided?
(3) The content task and structure task are measured in a decoupled manner. Are these tasks entirely independent, and theoretically, could the low-frequency information be used as the query set to predict within the raw image support set, and vice versa for high-frequency information? And so on.
(4) Besides the EMA (Exponential Moving Average) update method, what other update mechanisms have been considered or could be applicable to the model's training process?
(5) There seems to be a limited ablation study regarding the choice of loss functions. Why was MSE chosen for the Feature Reconstruction Loss, and could cosine similarity loss be a viable alternative? Similarly, why was KL divergence chosen for the Prediction Loss, and was this decision based on empirical results or theoretical analysis?
(6)The experiments were conducted using ResNet-10. Could the authors provide insights into whether the proposed method could be applied to larger models such as ResNet-101? Additionally, could the method be beneficial when applied to models like CLIP from OpenAI or the DINO model, and are there any explorations in this direction?
(7) Some of the experimental content in the supplementary materials could be incorporated into the main text, such as the sections on 'low' and 'high' in Table 4. This is just a suggestion.

**Limitations:**

The authors have transparently addressed the limitations of their method, particularly its performance on the Chest dataset, and have responsibly considered the broader societal impacts, confirming no negative social effects. They also acknowledge the increased training time associated with their approach. Proactively, they suggest future improvements and responsibly guide how their work can contribute to the field's progression, aligning well with the checklist for addressing limitations and societal impacts.

---

> ### Author Rebuttal · Authors · 2024-08-07
>
> ## To Reviewer mWuv :
> ### Weaknesses (1)
> ### Response :
>
> We present the computational costs during the training and inference phases in Table 2 and 3 (Due to character limitations, please see "To Reviewer SF8W".), respectively. We can draw the following observations.
>
> 1) During the training, the computational overhead of our method mainly comes from the backbone and image decomposition. As shown in Table 2 (please see "To Reviewer SF8W"), the backbone and the feature reconstruction network are lightweight, which to some extent mitigates the computational overhead.
>
> 2) During inference, our method does not introduce additional computational overhead, resulting in efficient inference and good generalization.
>
> ### Weaknesses (2)  &  Questions (5) & Questions (6)
>
> ### Response 1:  Using Resnet-101 and ViT as backbones.
>
> We validate our method with Resnet-101 and ViT as backbones, as shown in Table 6 (Please see Table 6 in the global rebuttal). It can be seen that our method effectively scales to different backbones and demonstrates certain performance advantages.
>
> ### Response 2: Verifying cosine loss.
>
> We validate cosine similarity as an alternative to MSE, as shown in Table 7 (Please see Table 7 in the global rebuttal). It can be observed that MSE and cosine similarity yield nearly equivalent performance.
>
> ### Response 3: About KL divergence.
>
> In our alignment prior, we use the predictions from the main branch as an anchor, and expect the high/low-frequency predictions to align with this anchor. To achieve this, we use KL divergence as the metric. The reasons are as follows. First, since the anchor is dynamically changing during training, meaning the entropy of the anchor is not constant, cross-entropy is not suitable. Second, our alignment loss is asymmetric, so JS divergence is also not applicable. In addition, Wasserstein-1 distance could serve as an alternative to KL divergence. However, it requires computing the optimal transport between different prediction distributions, which involves quadratic programming and results in higher computational complexity.
>
> ### Weaknesses (3)
> ### Response :
>
> Thanks for the suggestions. In this work, we resort to the image decomposition prior which have been proved shared across different images despite their domains, i.e., each image can be decomposed into complementary low-frequency content details and high-frequency robust structural characteristics. More importantly, we specially establish a feature reconstruction prior and a prediction consistency prior to separately encourage the consistency. This allows for collectively guiding the network’s meta-learning process with the aim of learning cross-domain generalizable embeddings.  We will supplement more theoretical analysis into the manuscript.
>
> ### Questions (1)
> ### Response :
>
> We validate the hyper-parameters in FFT(e.g., radius ratio), which determines the boundary between high and low frequencies. The results are shown in Table 8 (Please see Table 8 in the global rebuttal). It can be observed that this hyper-parameter does not significantly affect the overall performance. We set it to 0.5 for all cases.
>
>
> ### Questions (2)
> ### Response 1 :  About training epoch.
>
> We validate the impact of training epochs on performance. The results are shown in Table 9 (Please see Table 9 in the global rebuttal). It can be observed that as the number of epochs increases, performance initially rises and then declines. However, since we cannot link to the test data during the training phase, we can only judge based on the loss curve during training (please see Fig. 2 in Response.pdf). We observed that the loss tends to converge around 50 epochs. Therefore, we set it to 50 for all cases.
>
> ### Response 2 :  About loss curve and landscape.
>
> We provide the loss optimization curve and loss landscape in Fig. 2. As can be seen, the baseline tends to over-fit at the relatively early epochs. In contrast, with the assistance of prior regularization, our method can mitigate over-fitting to some extent. For the loss landscape, we followed the implementation in [1].  Firstly, randomly perturb the model trained in the source domain in 2000 directions; Secondly, perform inference on the target domain for each perturbed model, and record the loss value; Finally, we visualize the loss landscape based on the recorded loss values and directions. It can be seen that, compared to the baseline, the trained model of the proposed method is more robust to unknown perturbations. For example, the loss values in our method converge within a certain range. In contrast, the baseline method's loss values tend to diverge. This demonstrates the good generalization ability of our method.
>
> [1] Visualizing the loss landscape of neural nets. NIPS, 2018.
>
> ### Questions (3)
> ### Response :
>
> We designed a variant experiment that utilizes the raw support set to predict high-frequency and low-frequency query. The results are shown in Table 10 (Please see Table 10 in the global rebuttal). It can be seen that this variant also achieves significant improvements over the baseline. However, the performance of this variant is slightly lower than our original approach. This may be because independently using high-frequency or low-frequency support images to predict high-frequency or low-frequency query images can more explicitly utilize the diverse frequency information.
>
> ### Questions (4)
> ### Response :
>
> We supplement some experiments to verify alternative mechanisms, and the results are shown in Table 11 (Please see the global rebuttal). We compared three update mechanisms. Case 1: we keep parameters shared among the three branches. Case 2: non-shared parameters between the three branches and joint training in an end-to-end manner. Case 3 (Ours): EMA strategy to update. As can be seen, our method achieves the best performance.
>
> ### Questions  (7)
> ### Response :
>
> Thanks for the suggestions. We will supplement those results into the main text.

---

### Official Review · Reviewer_SF8W · 2024-07-11

**Soundness:** 3
**Presentation:** 3
**Contribution:** 3
**Rating:** 6
**Confidence:** 5

**Summary:**

The paper introduces a novel framework called Meta-Exploiting Frequency Prior for Cross-Domain Few-Shot Learning, which aims to improve meta-learning's generalization by decomposing images into high- and low-frequency components. This method leverages these components to guide the feature embedding network, enhancing category prediction consistency and feature reconstruction. The framework achieves state-of-the-art results on various cross-domain few-shot learning benchmarks, demonstrating its effectiveness and efficiency.

**Strengths:**

1. The paper presents a novel approach by introducing the concept of exploiting cross-domain invariant frequency priors for few-shot learning. The idea of using cross-domain invariant frequency priors is interesting.

2. The proposed framework is rigorously evaluated on multiple cross-domain few-shot learning benchmarks, demonstrating its effectiveness.

3. Good writing and clear presentation.

**Weaknesses:**

1. The method relies heavily on the strong prior assumption that using low and high-frequency components can effectively mimic the distribution shift encountered during testing. However, this assumption may not hold true when the test and training datasets are entirely unrelated, as observed in the EuroSAT dataset where the performance of the method is not as impressive. The authors need to clarify and justify this assumption.

2. The paper does not provide an evaluation of the method's performance on within-domain few-shot learning tasks. Understanding how the proposed framework performs in a scenario where the training and testing data come from the same domain could provide a more comprehensive view of its robustness and generalizability.

3. The computational complexity of the proposed framework during the training phase is not thoroughly discussed. While the paper claims no additional inference cost, the added steps of image decomposition and feature reconstruction during training could introduce significant computational overhead. Detailed analysis and discussion on the computational requirements and efficiency would be beneficial.

4. The impact of the choice of image decomposition method (e.g., Fast Fourier Transform) on the overall performance of the framework is not thoroughly explored. Alternative decomposition techniques may yield better results or be more computationally efficient. A comparison of different decomposition methods and their influence on the performance and efficiency of the framework would provide valuable insights and strengthen the paper's contributions.

**Questions:**

See weakness

**Limitations:**

yes

---

> ### Author Rebuttal · Authors · 2024-08-07
>
> ## To Reviewer SF8W:
> ### Weaknesses 1
> ### Response :
> We compared the FFT decomposition results of natural images and EuroSAT images (please see Fig. 1 in Response.pdf). We observed that FFT decomposition of natural images can obtain clear low-frequency and high-frequency information. However, the high-frequency part of the image from EuroSAT is almost all noise. There may be two reasons for this. One is that during the imaging process of remote sensing scene images, due to the high spatial resolution of remote sensing scene images, the camera often compresses high-frequency information to improve imaging efficiency. Second, remote sensing scene images focus on containing low-frequency texture content and lack clear high-frequency structural information.
>
> We will further clarify this weakness in the manuscript.
>
>
>
> ### Weaknesses 2
> ### Response :
> Following the suggestion, we have supplemented experiments for the proposed method in the in-domain setting. All in-domain experiments followed standard settings, the results as shown in Table 1. It can be seen that the proposed method achieved good results with both Resnet-10 and ViT-S backbones.
>
> We will supplement these experimental results into the manuscript.
>
>
> **Table 1 : Performance on within-domain few-shot learning tasks under 5-way 5-shot setting.**
> |　Method　|　　Backbone　|　mini-ImageNet　|　CIFAR-FS　|
> |---|---|---|---|
> |　ProtoNet[1]　|　ResNet-12 　|　  80.53% 　|　 83.5%±0.5% 　|
> |　MetaOptNet[2]　|　ResNet-12 　|　 78.63%±0.46% 　|　84.3%±0.5% 　|
> |　SetFeat[3]　|　ResNet-12 　|　 82.71%±0.46% 　|　- 　|
> |　DiffKendall[4]　|　ResNet-12 　|　 80.79%±0.31% 　|　- 　|
> |　MetaDiff[5]　|　ResNet-12 　|　 81.21%±0.56% 　|　- 　|
> |　**Ours**　|　ResNet-10 　|　 83.30%±0.57% 　|　86.86%±0.63% 　|
>
> |　Method　|　　Backbone　|　mini-ImageNet　|　CIFAR-FS　|
> |---|---|---|---|
> |　P>M>F[6]　|　ViT-S (DINO pretrain) 　|　  98.0% 　|　92.5%   |　- 　|
> |　**Ours**　|　ViT-S (DINO pretrain)　|　 98.78%±0.12% 　|　93.75%± 0.10% 　|
>
> [1] Prototypical networks for few-shot learning， NIPS 2017.
>
> [2] Meta-learning with differentiable convex optimization，CVPR 2019.
>
> [3] Matching Feature Sets for Few-Shot Image Classification, CVPR 2022.
>
> [4] DiffKendall: A Novel Approach for Few-Shot Learning with Differentiable Kendall's Rank Correlation, NIPS 2023.
>
> [5] MetaDiff: Meta-Learning with Conditional Diffusion for Few-Shot Learning, AAAI 2024.
>
> [6] Pushing the Limits of Simple Pipelines for Few-Shot Learning: External Data and Fine-Tuning Make a Difference, CVPR 2022.
>
> ### Weaknesses 3
> ### Response :
>
> Following the suggestion, we present the computational costs of the proposed method during the training and inference phases in Tables 2 and 3, respectively. We can draw the following conclusions.
>
> 1) The backbone network and the feature reconstruction network in our method are lightweight, which to some extent mitigates the computational overhead.
>
> 2) In addition, a portion of the computational overhead of our method comes from the FFT. In future work, we will explore the use of efficient, learnable image decomposition techniques, employing differentiable convolutional kernels to capture information from different frequency bands of the images.
>
> 3) During the inference phase, the proposed method does not introduce additional computational overhead, resulting in efficient inference and good generalization.
>
> We will supplement these analyses and discussion into the manuscript.
>
> **Table 2 : About  parameters (M), FLOPs (G), and Iteration time (S) during training phase. Take the 5-way 1-shot 15-query task as an example to calculate the floating point and training iteration time. N represents the number of pixels in each image.**
> |　Items　|　Decomposition　|　Backbone 　|　Reconstruction 　|
> |---|---|---|---|
> |　Parameters　|　0　|　4.9057　|　0.1313　|
> |　FLOPs　|　O(NlogN) |　71.6513　|　0.0105　|
> |　Iteration time　|　0.6912 |　1.3859　|　0.1039　|
>
>
> **Table 3 : Performance and efficiency during target domain inference phase under 5-way 1-shot (5-way 5-shot ) setting.**
> |　Method　|　Inference time　|　Average performance 　|
> |---|---|---|
> |　Baseline　|　0.06 (0.07) 　|　51.90 (71.77)　|
> |　Ours　|　0.06 (0.07) |　54.16 (74.87)　|
>
> ### Weaknesses 4
> ### Response :
>
> Following the suggestion, we compared the performance and efficiency under different decomposition methods. The results are shown in Tables 4 and 5. It can be seen that, compared to the baseline methods, the proposed method achieves better results across various decomposition methods. Additionally, using wavelet-based decomposition results in higher computational efficiency.
> When we prioritize efficiency, wavelet decomposition is the optimal choice. However, compared to FFT, the performance of wavelet decomposition is slightly lower. Therefore, overall, FFT remains a better choice.
>
> We will supplement these analyses and discussion into the manuscript.
>
> **Table 4 : Performance on different image decomposition method under 5-way 1-shot (5-way 5-shot ) setting.**
> |　Method　|　CUB　|　Places　|　Plantae　|　CropDisease　|　Ave.　|
> |---|---|---|---|---|---|
> |　Baseline　|　47.05 (67.99) |　51.09 (71.74) 　|　39.26 (57.82) 　| 　 70.22 (89.54) 　|　 51.90 (71.77) 　|
> |　Haar-wavelet　|　49.12 (71.12 ) 　|　52.63 (73.92)|　40.56 (60.46)　| 　70.82 (90.45) 　|　53.28 (73.98)　|
> |　DB-wavelet　|　49.52 (71.59) 　|　52.74 (73.65)|　40.78 (60.60)　| 　70.87 (90.54) 　|　53.47 (74.09)　|
> |　FFT　|　51.55 (73.61 ) 　|　52.06 (73.78)|　41.55 (61.39)　| 　71.47 (90.68)|　54.16 (74.87)　|
>
>
> **Table 5 : Decomposition efficiency (sec).**
> |　Method　|　One image　|　All images in 5-way 1-shot 15-query task 　|
> |---|---|---|
> |　Haar-wavelet　|　0.0015　|　0.1200　|
> |　DB4-wavelet　|　0.0016 |　0.1280　|
> |　FFT　|　0.0086 |　0.6912　|

---

### Author Rebuttal · Authors · 2024-08-07

Author Response for ``Meta-Exploiting Frequency Prior for Cross-Domain Few-Shot Learning''

We would like to express our gratitude to the AC and the reviewers for their valuable comments and suggestions. Over the past week, we have responded to all comments mentioned by the reviewers. The response includes:
1) evaluating the computational overhead during the training phase.
2) clarifying the prior assumptions and theoretical advantages of the proposed method.
3) supplementing the experiments in the in-domain setting.
4) validating different image decomposition methods, hyper-parameters in FFT, training epochs, the loss optimization curve and loss landscape.
5) supplementing experiments with different backbone networks, such as Resnet-101 and ViT.
6) validating different update strategies, the choice of loss functions, and variants using the original images for prediction.
7) verifying the performance of the proposed method with further fine-tuning, etc.

For detailed responses, please refer to each reviewer's rebuttal window. Additionally, since we cannot upload figures in the reviewer's rebuttal window, we have included all figures in a separate "Response.pdf" within this global rebuttal window. Besides, due to character limitations in the reviewer's window, we have placed some experimental result tables in this global rebuttal window.


**Table 6 : Performance of the proposed method on different backbone networks under 5-way setting. The results of ProtoNet[1] are derived from our reproduction, and the results  of P>M>F[2] were copied from its paper.**
|Method|Backbone|Setting|CUB|Cars|Places|Plantae|Chest|ISIC|EuroSAT|CropDisease|Ave.|
|---|---|---|---|---|---|---|---|---|---|---|---|
|　ProtoNet[1]　|　Resnet101　|　1-shot　|　52.14　|　35.08 　|　52.40 　|　39.01　|　22.01　|　33.34　|　62.30　|　68.34　|　45.57　|
|　**Ours**　|　Resnet101　|　1-shot　|　53.98　|　37.95　|　52.06　|　42.44　|　22.32　|　36.10　|　62.54　|　71.74　|　47.39　|
|　ProtoNet[1]　|　Resnet101　|　5-shot　|　75.05　|　51.48　|　73.29　|　58.41　|　26.18　|　47.96　|　78.09　|　88.96　|　62.42　|
|　**Ours**　|　Resnet101　|　5-shot　|　76.87　|　56.80　|　74.61　|　61.64　|　26.44　|　50.20　|　80.17　|　90.57　|　64.66　|

|Method|Backbone|Setting|CUB|Cars|Places|Plantae|Chest|ISIC|EuroSAT|CropDisease|Ave.|
|---|---|---|---|---|---|---|---|---|---|---|---|
|　P>M>F[2]　|　ViT-S (DINO pretrain)　|　1-shot　|　78.13　|　37.24　|　71.11　|　53.60　|　21.73　|　30.36　|　70.74　|　80.79　|　55.46　|
|　**Ours**　|　ViT-S (DINO pretrain)　|　1-shot　|　85.07　|　42.82　|　73.71　|　56.6　|　23.32　|　35.13　|　72.03　|　81.82　|　58.81　|
|　P>M>F[2]　|　ViT-S (DINO pretrain)　|　5-shot　|　-　|　-　|　-　|　-　|　27.27　|　50.12　|　85.98　|　92.96　|　-　|
|　**Ours**　|　ViT-S (DINO pretrain)　|　5-shot　|　97.5　|　74.6　|　89.24　|　81.63　|　26.31　|　51.72　|　89.95　|　95.93　|　75.86　|

[1] Prototypical networks for few-shot learning，NIPS 2017.

[2] Pushing the Limits of Simple Pipelines for Few-Shot Learning: External Data and Fine-Tuning Make a Difference, CVPR 2022.



**Table 7 : Verifying cosine loss under 5-way 1-shot (5-shot) setting.**
|　Method　|　CUB　|　Places　|　Plantae　|　CropDisease　|　Ave.　|
|---|---|---|---|---|---|
|　Cosine similarity loss　|　51.50 (73.54) |　52.08 (73.78) 　|　41.54 (61.40) 　| 　 71.49 (90.66) 　|　 54.15 (74.84) 　|
|　MSE　loss　|　51.55 (73.61 ) 　|　52.06 (73.78)　|　41.55 (61.39)　| 　71.47 (90.68)|　54.16 (74.87)　|




**Table 8 : Verifying the hyper-parameters in FFT under 5-way 1-shot (5-shot ) setting.**
|　Radius_ratio　|　CUB　|　Places　|　Plantae　|　CropDisease　|
|---|---|---|---|---|
|　0.1　|　51.14 (73.25)　|　51.95 (73.76) 　|　41.40 (61.33) 　| 　 71.68 (90.89) 　|
|　0.3　|　51.27 (73.33)　|　52.18 (73.81) 　|　41.47 (61.44) 　| 　 71.67 (90.83) 　|
|　**0.5**　|　51.55 (73.61)　|　52.06 (73.78)　 |　41.55 (61.39)　 | 　 71.47 (90.68)　 |
|　0.7　|　51.30 (73.35)　|　51.89 (73.79) 　|　41.33 (61.28) 　| 　 71.37 (90.61) 　|
|　0.9　|　51.23 (73.10)　|　52.00 (73.85) 　|　41.30 (61.27) 　| 　 71.54 (90.63) 　|


**Table 9 : Verifying the training epoch under 5-way 1-shot (5-shot ) setting.**
|　Epoch　|　CUB　|　Places　|　Plantae　|　CropDisease　|
|---|---|---|---|---|
|　40　|　51.12 (72.88)　|　52.53 (74.04) 　|　41.04 (60.71) 　| 　 71.15 (90.56) 　|
|　**50**　|　51.55 (73.61)　|　52.06 (73.78)　 |　41.55 (61.39)　 | 　 71.47 (90.68)　 |
|　60　|　51.00 (73.45)　|　51.60 (73.59) 　|　41.39 (61.06) 　| 　 71.17 (90.33) 　|
|　80　|　50.69 (72.80)　|　51.87 (73.34) 　|　40.88 (60.87) 　| 　 70.75 (90.27) 　|
|　100　|　50.02 (72.12)　|　51.44 (73.15) 　|　40.91 (60.98) 　| 　 70.36 (90.19) 　|


**Table 10 : Verifying the variant that predict within the raw image support set under 5-way 1-shot (5-shot ) setting.**
|　Method　|　CUB　|　Places　|　Plantae　|　CropDisease　|　Ave.　|
|---|---|---|---|---|---|
|　Baseline　|　47.05 (67.99) |　51.09 (71.74) 　|　39.26 (57.82) 　| 　 70.22 (89.54) 　|　 51.90 (71.77) 　|
|　Variant　|　51.32 (73.18) 　|　52.12 (73.87) 　|　41.61 (61.10)　| 　70.91 (90.19) 　|　53.99 (74.58)　|
|　Ours　|　51.55 (73.61 ) 　|　52.06 (73.78) 　|　41.55 (61.39)　| 　71.47 (90.68) 　|　54.16 (74.87)　|



**Table 11 : Comparing alternative update mechanisms under 5-way 1-shot (5-shot) setting.**
|　Method　|　CUB　|　Places　|　Plantae　|　CropDisease　|　Ave.　|
|---|---|---|---|---|---|
|　Baseline　|　47.05 (67.99) |　51.09 (71.74) 　|　39.26 (57.82) 　| 　 70.22 (89.54) 　|　 51.90 (71.77) 　|
|　Case 1　|　50.76 (72.56) |　51.83 (72.76) 　|　40.73 (60.46) 　| 　69.47 (89.56) 　|　53.19 (73.83) 　|
|　Case 2　|　50.93 (72.59) | 　52.36 (73.84) | 　41.00 (60.74) |　 70.65 (90.40)　|　53.73 (74.39) |
|　Case 2 (ours)　|　51.55 (73.61 ) 　|　52.06 (73.78)|　41.55 (61.39)　| 　71.47 (90.68)　|　54.16 (74.87)　|

---

### Decision · Program_Chairs · 2024-09-25

**Decision:**

Accept (poster)

**Comment:**

In this paper, the authors propose to improve cross-domain few-shot learning by decomposing images into high- and low-frequency components. Extensive experimental results are provided to show the effectiveness of the proposed method. After the rebuttal period, this paper receives consistent positive ratings. Therefore, it is a clear acceptance.